# Redox controls on methane formation, migration and fate in shallow aquifers

Pauline Humez[1], Bernhard Mayer[1], Michael Nightingale[1], Veith Becker[1], Andrew Kingston[1], Stephen Taylor[1], Guy Bayegnak[2], Romain Millot[3] and Wolfram Kloppmann[3]

[1]Applied Geochemistry Group, Department of Geoscience, University of Calgary, 2500 University Drive NW, Calgary, Alberta, Canada T2N 1N4
[2]Alberta Environmental Monitoring, Evaluation and Reporting Agency (AEMERA), 9th Floor 9888 Jasper Avenue, Edmonton, Alberta, Canada T5J 5C6
[3]BRGM, French Geological Survey, 3 avenue Claude Guillemin, BP 6009, 45060 Orléans Cedex 2, France

*Correspondence to*: P. Humez (phumez@ucalgary.ca)

**Abstract.** Development of unconventional energy resources such as shale gas and coalbed methane has generated some public concern with regard to the protection of groundwater and surface water resources from leakage of stray gas from the deep subsurface. In terms of environmental impact to and risk assessment of shallow groundwater resources, the ultimate challenge is to distinguish: (a) natural in-situ production of biogenic methane, (b) biogenic or thermogenic methane migration into shallow aquifers due to natural causes, and (c) thermogenic methane migration from deep sources due to human activities associated with the exploitation of conventional or unconventional oil and gas resources. This study combines aqueous and gas (dissolved and free) geochemical and isotope data from 372 groundwater samples obtained from 186 monitoring wells of the provincial Groundwater Observation Well Network (GOWN) in Alberta (Canada), a province with a long record of conventional and unconventional hydrocarbon exploration. We investigated whether methane occurring in shallow groundwater formed in-situ, or whether it migrated into the shallow aquifers from elsewhere in the stratigraphic column. It was found that methane is ubiquitous in groundwater in Alberta and is predominantly of biogenic origin. The highest concentrations of biogenic methane (> 0.01 mM or > 0.2 mg/L), characterized by $\delta^{13}C_{CH4}$ values <–55 ‰, occurred in anoxic Na-Cl, Na-$HCO_3$ and Na-$HCO_3$-Cl type groundwaters with negligible concentrations of nitrate and sulfate suggesting that methane was formed in-situ under methanogenic conditions for 39.1% of the samples. In only a few cases (3.7 %) was methane of biogenic origin found in more oxidizing shallow aquifer portions suggesting limited upward migration from deeper methanogenic aquifers. 14.1% of the samples contained methane with $\delta^{13}C_{CH4}$ values > –54 ‰, potentially suggesting a thermogenic origin, but aqueous and isotope geochemistry data revealed that the elevated $\delta^{13}C_{CH4}$ values were caused by microbial oxidation of biogenic methane or post-sampling degradation of low $CH_4$ content samples rather than migration of deep thermogenic gas. A significant number of samples (39.2%) contained methane with predominantly biogenic C isotope ratios ($\delta^{13}C_{CH4}$ <–55 ‰) accompanied by elevated concentrations of ethane and sometimes trace concentrations of propane. These gases observed in 28.1% of the samples, bearing both biogenic ($\delta^{13}C$) and thermogenic (presence of $C_3$)

characteristics, are most likely derived from shallow coal seams that are prevalent in the Cretaceous Horseshoe Canyon and neighboring formations in which some of the groundwater wells are completed. The remaining 3.7% of samples were not assigned because of conflicting parameters in the datasets or between replicates samples. Hence, despite quite variable gas concentrations and a wide range of $\delta^{13}C_{CH4}$ values in baseline groundwater samples, we found no conclusive evidence for deep thermogenic gas migration into shallow aquifers either naturally or via anthropogenically-induced pathways in this baseline groundwater survey. This study shows that the combined interpretation of aqueous geochemistry data in concert with chemical and isotopic compositions of dissolved and/or free gas can yield unprecedented insights into formation and potential migration of methane in shallow groundwater. This enables the assessment of cross-formational methane migration and provides an understanding of alkane gas sources and pathways necessary for a stringent baseline definition in the context of current and future unconventional hydrocarbon exploration and exploitation.

**Keywords.** Methane • Alberta • groundwater • stable isotope • geochemistry • redox processes

## 1 Introduction

Development of unconventional energy resources such as shale gas and coalbed methane is often accompanied by concerns of some landowners and parts of the public that shallow groundwater could be affected by leakage of stray gas from the deep subsurface. To address unambiguously such concerns, it is essential to assess the natural occurrence of methane and its spatial distribution, the variability of methane concentrations and the sources of methane in shallow groundwater prior to unconventional energy development to establish a baseline. In the last 5 years, an increasing number of publications have addressed the questions of occurrence and sources of methane in shallow groundwater in natural gas producing regions (Osborn et al., 2011a and b; Warner et al., 2013; Darrah et al., 2012, 2014; Jackson et al., 2013; Molofsky et al., 2013; Siegel et al., 2015; Vengosh et al., 2013; Brantley et al., 2014; Baldassare et al., 2014; McPhilips et al., 2014; McInstosh et al., 2014; Vidic et al., 2013). These studies have contributed a wealth of baseline data for gas occurrences in shallow groundwater in many regions of North America. Some of these studies reported that elevated methane concentrations in shallow aquifers were correlated with geology, especially the occurrence of low-sulfur coal deposits, and topography, since groundwater from wells in valleys tended to have higher methane concentrations (Mathes and White, 2006; Molofsky et al., 2013, Etiope et al., 2013). In other cases, methane concentrations were correlated with groundwater types with elevated methane concentrations predominantly reported in sodium chloride or sodium bicarbonate groundwater types (Molosky et al., 2013; McPhilips et al., 2014). Where thermogenic gas was found in shallow groundwater, it is however not always clear to what extent this occurred inadvertently as a result of human activities or due to natural flowpaths.

Assessment of the aqueous geochemistry and the redox conditions in the aquifers affected by elevated methane concentrations can reveal whether methane formed in-situ, or whether it formed elsewhere in the stratigraphic column and migrated into the shallow aquifer. In terms of potential environmental impact and risk assessment focusing on shallow

groundwater resources, the ultimate challenge is to distinguish: (a) natural in-situ production of biogenic methane in methanogenic aquifers, (b) biogenic or thermogenic methane naturally migrating into shallow aquifers, and (c) predominantly thermogenic methane from deep sources migrating due to human activities associated with exploitation of conventional or unconventional oil and gas resources.

In-situ formation of methane in shallow aquifers requires highly reducing conditions. According to the 'redox ladder' concept, microbial formation of $CH_4$ can only occur after dissolved oxygen is consumed, denitrification has removed nitrate, and bacterial sulfate reduction has progressed towards completion (Appelo and Postma, 2005; Barker and Fritz, 1981; Darling and Goody, 2006; Whiticar, 1986). Hence, analyzing a variety of water chemistry parameters (e.g. the redox couples $Fe^{3+}/Fe^{2+}$, $NO_3^-/NO_2^-$, $CO_2/CH_4$, $SO_4^{2-}/H_2S$) can provide important clues to whether in-situ formation of methane within a shallow aquifer is possible. In addition, biogenic methane formed in shallow aquifers is characterized by very negative $\delta^{13}C$ and $\delta^2H$ values (e.g. Whiticar, 1999). Therefore, isotope analyses on methane, higher alkanes (where present) and other dissolved groundwater constituents such as dissolved inorganic carbon ($\delta^{13}C_{DIC}$), nitrate ($\delta^{15}N_{NO3}$, $\delta^{18}O_{NO3}$) and sulfate ($\delta^{34}S_{SO4}$, $\delta^{18}O_{SO4}$) can provide important additional insights about redox conditions and methane formation pathways in shallow aquifers.

Combined geochemical and isotopic analyses on groundwater and its dissolved or free gases phases thus have the potential to determine whether in-situ methane formation is possible or if gas migration must have occurred. The latter would be for instance the case if biogenic methane is found in aerobic or non-methanogenic aquifer sections. Alternatively, if thermogenic methane with elevated $\delta^{13}C$ values accompanied by ethane and propane (e.g. Whiticar, 1999) is detected in shallow aquifers, gas migration from deeper geological formations into shallow aquifers must be postulated. In these cases, it is desirable to determine the depth of the gas source and its natural or anthropogenic migration pathways. It is equally important to identify apparent (or pseudo-) thermogenic methane characterized by elevated $\delta^{13}C$ values that are in reality caused by microbial oxidation of biogenic methane enriching the remaining methane in $^{13}C$ (Barker and Fritz, 1981), rather than by migration of thermogenic gas from deep geological sources.

Alberta is a province in Western Canada with a long history of conventional and unconventional energy exploitation. Conventional oil and natural gas have been produced from numerous reservoirs in the province since the first natural gas find in 1883 and more than 400,000 oil and natural gas wells have been drilled (e.g. Breen, 1993). More recently, unconventional natural gas has been exploited from rather shallow coalbed deposits (200-800 m below ground), predominantly in the Horseshoe Canyon Formation in and east of the Edmonton to Calgary corridor (ECC) in the southeastern part of the province with a peak activity occurring between 2006 and 2010. In the last decade, shale gas exploration and exploitation have also commenced in the Triassic Montney and the Devonian Duvernay Formations in the northwestern part of the province typically at depths exceeding 1.5 km. Therefore, assessment of the occurrence and the sources of methane in shallow groundwater is of key importance for two reasons: 1) to assess whether previous oil and gas exploitation has caused negative impacts on shallow groundwater due to stray gas contamination; and 2) to establish a

baseline against which potential future impacts of stray gas migration on shallow aquifers, or the lack thereof, can be determined.

A baseline study was conducted between 2006 and 2014 investigating the occurrence of methane in shallow groundwater of Alberta (Canada) obtained from provincial monitoring wells. The objective was to determine, based on comprehensive aqueous geochemical and isotopic data evaluations of groundwater samples and their dissolved and free gases, the distribution and sources of methane in shallow groundwater. An additional goal was to characterize the hydrochemical environment in which methane was formed or transformed through redox processes to evaluate whether methane occurring in shallow groundwater in Alberta at baseline conditions has formed in-situ and under which geochemical conditions, or whether methane had formed elsewhere and has migrated into the shallow aquifers.

## 2 Background: Materials and Methods

### 2.1 Study site GOWN network

The Groundwater Observation Well Network (GOWN) of the Alberta Government initiated in 1955 (Alberta Research Council, 1956) and taken over by Alberta Environment (AENV) in 1982 is comprised of groundwater monitoring wells completed in various shallow aquifers throughout the province (Fig. 1). A recent comprehensive monitoring program collects water level information together with geochemical and isotopic data (H, O, C, S) since 2006 in order to record potential impacts on quantity and quality of groundwater in Alberta. The GOWN consists currently of over 250 active observation wells with many wells located in coalbed methane (CBM) production areas in the southeastern part of the province, while few wells exist in the shale gas development regions in the northwest as shown by the unequal spatial distribution of the well locations in Fig. 1. Since 2006, groundwater samples and dissolved gas and free gas samples from GOWN wells have been routinely obtained where possible for chemical and isotopic analyses. A first assessment of the gas geochemical dataset has been reported by Humez et al. (2015, 2016). This study evaluates both aqueous and gas geochemical and isotopic data of the GOWN monitoring program of shallow groundwater samples. GOWN wells are drilled into aquifers either within surficial deposits from the last major glaciation or reach sedimentary bedrock of usually Paleogene or Cretaceous age. The depth of the GOWN wells varies from 26 m to 250 m with an average of 60 m below ground surface (bgs) accessing groundwaters from different shallow aquifers (Fig. 1). The wells have typically stainless steel casing with diameters varying from 32 mm to 254 mm with an average of 109 mm and they typically have short stainless steel or PVC screens only in the target aquifer formation.

The aquifer lithologies vary considerably comprising mostly fractured mudstone, sandstone and siltstone beds or lenses, pre-glacial sand, or surficial sandy and gravelly lacustrine or moraine deposits (Dawson et al., 1994, Fig. 1). The regional Upper Cretaceous-Paleogene stratigraphy differentiates many sandy clastic depositions including the (i) Lower Campanian Milk River Formation (and equivalents), (ii) Middle to Upper Campanian Belly River (Judith River) Group (and equivalents), (iii) Upper Campanian to Lower Maastrichtian Horseshoe Canyon Formation (and equivalents), (iv) Upper Maastrichtian to

Lower Paleocene Scollard Formation (and equivalents), and (v) the Middle to Upper Paleocene Paskapoo Formation (and equivalents), which comprise the major shallow aquifers in the study area. Four sedimentary units with more fine-grained materials comprise the (i) the Lower Campanian Pakowki Formation (upper Lea Park Formation), (ii) the Middle Campanian Bearpaw shales, (iii) the Maastrichtian Battle Shales Formation, and (iv) the upper part of the Scollard Formation (Dawson et al., 1994), which are typically classified as aquitards (Fig. 1).

The Quaternary deposits include the Muriel Lake Formation that is composed of silt, sand and gravel of glacio-fluvial origin. Among the stratigraphic intervals containing coal zones with CBM potential are the Lower Cretaceous Mannville Group (e.g. Mannville Group coals), and the Belly River (e.g. McKay Coal, Taber Coal, Lethbridge Coal zones), Horseshoe Canyon (e.g. Drumheller Coal zone, a primary CBM target), and Scollard Formations (e.g. Ardley Coal zone). Thin coal seams occur also throughout the Paskapoo Formation. More information about these geological formations can be found in Meyboom (1960), Rosenthal et al. (1984), Hamblin (1998), Hamblin (2004), Dawson et al. (1994), Lyster and Andriashek (2012), Grasby et al. (2008), and Prior et al. (2013).

### 2.2 Samples and laboratory techniques

Between 2006 and 2014, a total of 372 groundwater samples were obtained from 186 GOWN wells accessing various shallow aquifers throughout Alberta. Many wells were sampled repeatedly, either on the same day as replicates for sampling and analytical quality control, or at greater time intervals to assess temporal water quality variations. All samples, including the replicates, are considered as individual samples in this paper. The database contains both aqueous and gaseous geochemical data for 372 samples. An electrical charge balance criteria for cations and anions of ± 10 % was applied for the aqueous geochemical data. Since calcium concentrations were not reported for a number of samples 35% of the groundwater samples were excluded leaving 242 samples (criteria #1, Table 1). Eight additional samples were discarded because of no gas composition data were reported (criteria #2, Table 1). Among the 234 remaining samples, 150 samples had dissolved and free gas information, and 80 samples had only dissolved gas analyses reported. Four samples had only information on free gas without dissolved gas concentrations being reported. Among these 234 samples, 9 samples had no chemical data reported. Hence, a total of 225 samples have been evaluated as they have information on gas composition associated with balanced major ion chemistry i.e. Ca, Mg, Na, K, $SO_4$, $NO_3$, Cl, DIC (dissolved inorganic carbon) required to evaluate the water type of the investigated samples (criteria #3, Table 1). 135 of 225 samples (60%) contained ion chemistry, gas analyses and carbon isotopic data for methane in free (n= 100) or dissolved (n= 2) gas phases or both (n= 33) (Table 1).

#### 2.2.1 Major and Minor Ion analysis

To collect samples representative of aquifer conditions, the groundwater wells were purged until the field parameters pH, redox, dissolved oxygen, temperature, and electrical conductivity stabilized. Alberta Innovates Technology Futures (AITF) conducted the major and minor ion chemistry analyses on filtered samples (0.45µm) that were acidified to pH < 2 for cation

analysis and non-acidified for major anion determination. ICP-MS analysis was used to determine cation concentrations while titration for alkalinity and ion chromatography were used to determine anion concentrations. The detection limits are indicated in Table 2 and concentrations are expressed in mol/L (M).

### 2.2.2 Gas composition

A detailed description of the sampling equipment and procedures for free and dissolved gas samples is given in Humez et al. (2015). The composition of free and dissolved gas samples was determined in the laboratory by gas chromatography yielding concentrations for oxygen, carbon dioxide, methane and higher alkane chain compounds (such as ethane) with measurements conducted by AITF with uncertainties of ± 5% of the analytes. Gas composition data for free gas samples is reported in parts per million by volume (ppmv) and for dissolved gases expressed in mol/L (M) or mg/L to ensure comparability with other studies. The gas dryness parameter defined as the ratio between methane / higher n-alkanes was also determined.

### 2.3 Isotopic analyses

$\delta^{13}C_{CH4}$, $\delta^{2}H_{CH4}$, $\delta^{13}C_{DIC}$, $\delta^{34}S_{SO4}$, $\delta^{18}O_{SO4}$, $\delta^{15}N_{NO3}$, $\delta^{18}O_{NO3}$, $\delta^{2}H_{H2O}$ and $\delta^{18}O_{H2O}$ values were analyzed in the Isotope Science Laboratory at the University of Calgary. Stable isotope ratios are reported in the internationally accepted delta notation (‰) relative to VPDB for $\delta^{13}C$ values, VSMOW for $\delta^{2}H$ and $\delta^{18}O$ values, VCDT for $\delta^{34}S$ values and $N_2$ in air for $\delta^{15}N$ values. All carbon and hydrogen isotope analyses on methane and $CO_2$ were conducted on a ThermoFisher MAT 253 isotope ratio mass spectrometer (IRMS) coupled to a Trace GC Ultra + GC Isolink (ThermoFisher). The precision for carbon isotope analyses was better than ± 0.5‰ for hydrocarbons and better than ± 0.2‰ for carbon dioxide. The precision for hydrogen isotope analysis of hydrocarbons was better than 3‰. Water isotope analyses were performed by Off-axis Cavity Ringdown Spectroscopy using a Los Gatos Water Isotope Analyzer (DLT-100). Precision was better than ± 2 ‰ for $\delta^{2}H$ and ± 0.2 ‰ for $\delta^{18}O$.

To determine the isotopic composition of sulfate, dissolved sulfate was converted to barium sulfate ($BaSO_4$) and subsequently analyzed using a ThermoQuest Finnigan Delta$^{plus}$XL IRMS coupled with either a Fisons NA 1500 Elemental Analyzer for $\delta^{34}S_{SO4}$ analysis or a HEKAtech HT Oxygen Analyser with Zero-blank autosampler for $\delta^{18}O_{SO4}$ analysis. Precision for $\delta^{34}S_{SO4}$ and $\delta^{18}O_{SO4}$ is ± 0.5 ‰.

The isotopic composition of nitrate was determined on $N_2O$ generated by the denitrifier technique (c.f. Silva et al., 2000; Sigman et al., 2001; Casciotti et al., 2002) using a Thermo Delta V Plus IRMS coupled with a Finnigan MAT PreCon. Precisions of $\delta^{15}N_{NO3}$ and $\delta^{18}O_{NO3}$ are ± 0.3 ‰ and ± 0.7 ‰ respectively.

The analytical results for all groundwater samples were further investigated using PHREEQC (Parkurst and Appelo, 1999) to assess geochemical speciations, potential redox values (pe), ionic balance (< ± 10 %), among others. SPSS 22 was used for determining descriptive statistics such as median, mean, range, standard deviation and to evaluate the correlation between variables. Pearson correlation analysis was conducted where linear trends between two variables existed. When nonlinear

relationship between two variables existed or in presence of outliers, Spearman's rho and Kendall's tau tests were used instead (Humez et al., 2016).

## 3 Results

### 3.1 Field Parameters

During sampling of shallow groundwater in the field, temperature, electrical conductivity, dissolved oxygen content and oxidation-reduction potential (ORP) were determined for all water samples. The average groundwater temperature was 7 ± 3 °C, while the average pH value was 7.8. The electrical conductivity of the groundwater samples ranged from 212 to >16,000 µS/cm with an average value of 1634 µS/cm. Fourteen groundwater samples had dissolved oxygen concentrations > 0.06 mM (> 2 mg/L), 56 water samples had dissolved oxygen concentrations ranging between 0.01 and 0.06 mM (0.5 and 2.0 mg/L), and 129 samples had dissolved oxygen concentrations <0.01 mM (<0.5 mg/L). For water samples with redox potential reported, 70 samples had Eh values < 0 mV (Eh = 0.059*pe (Volt) with pe = log(e-)).

### 3.2 Major ion concentrations and hydrochemical water type classification

Major ion chemistry of groundwater and methane concentrations were determined for 225 groundwater samples from shallow aquifers and results are summarized in Table 2. Chloride concentrations ranged from 0.01 to 68.01 mM with mean and median values of 2.81 mM and 0.40 mM (n=225). Sulfate concentrations ranged from 4.55 µM to 74.16 mM with mean and median values of 2.83 mM and 0.72 mM (n=225) respectively. DIC concentrations ranged from 0.81 mM to 39.07 mM with mean and median values of 12.94 and 12.29 mM (n= 225) respectively. Nitrate concentrations ranged from 0.21 µM to 21.2 mM (n=136). For 90 samples, $NO_3$ concentrations have not been reported but the ion balance is acceptable so that $NO_3$ represents <<10 % of the anions.

The major cations Na, Ca, Mg, K showed a wide range of concentrations (Table 2). Sodium concentrations ranged from 0.03 mM to 165.78 mM with mean and median values of 17.28 mM and 12.76 mM (n=225) respectively. Calcium concentrations ranged from 0.01 mM to 9.08 mM with mean and median values of 0.94 mM and 0.40 mM (n=136) respectively. For 89 samples, Ca concentrations were not measured but the ion balance is acceptable so that Ca represents <10% of the cations (set to 0 for the Piper plot). Magnesium concentrations ranged from 2.26 µM to 12.45 mM with mean and median values of 0.60 mM and 0.05 mM (n=225) respectively. Potassium concentrations ranged from 0.01 mM to 0.54 mM with mean and median values of 0.07 mM and 0.05M (n=225) respectively. Total dissolved solids (TDS) were calculated by adding major ion concentrations. Total dissolved solids (TDS) ranged from 180 to 15,500 mg/L with an average value of 1264 mg/L.

The Piper plot in Fig. 2 shows that water types were found to be highly variable ranging from Ca-Mg-$HCO_3$ to Na-Cl types with the following order Na-$HCO_3$ (59.1%) > Na-$HCO_3$-Cl (17.0%) > Ca-$HCO_3$ (8.0%) > Na-Cl (5.8%) > Ca-Na-$HCO_3$ (5.3%) > Ca-$HCO_3$-Cl (2.2%) > Ca-Na-$HCO_3$-Cl (1.3%) > Ca-Na-Cl (0.9%) > Ca-Cl (0.4%). Elevated concentrations of

methane in groundwater were found predominantly in Na-Cl, $Na-HCO_3$ and $Na-HCO_3$-Cl water types (see color code in Fig. 2).

**3.3 Methane, ethane and propane occurrence in shallow groundwater**

Methane was detected above the limit of detection (DL) in all samples for which free gas analyses were available. The average methane concentration was 265,466 ppmv (n= 147). Twenty-five percent of the samples had methane concentrations > 390,000 ppmv (third quartile, $Q_3$). In dissolved gas samples, the average methane concentration was 0.43 mM (n=221). Twenty-five percent of the samples had a dissolved methane concentration >0.4 mM (third quartile, $Q_3$). The highest methane concentration was 3.01 mM (Table 2).

In free gas samples, the average ethane ($C_2H_6$) concentration was 215 ppmv (n=96) with a maximum value of 3650 ppmv. In dissolved gas samples, the average ethane concentration was 0.60 µM with a maximum of 17.63 µM. In free gas samples, the average detected propane ($C_3H_8$) concentration was 0.67 ppmv (n=36) with a maximum value of 4.60 ppmv. In dissolved gas samples, the average propane concentration was 0.03 µM with a maximum of 0.90 µM (Table 2).

Figure 3a shows that elevated dissolved methane concentrations were generally found at redox potentials (Eh) below 0 mV. Dissolved $CH_4$ concentrations and Eh values are weakly inversely correlated (Kendall's *tau* = -0.106 and $p < 0.05$, Spearman's *rho* = -0.167 and $p < 0.05$). One triplicate sample from a well located between Calgary and Red Deer had an elevated pe value while the methane concentrations were > 0.5 mM (Fig 3a). The highest methane concentrations in dissolved gas samples occurred at pH values > 7 and at low Eh < 0 mV (Fig. 3b).

A cross-plot of average TDS contents versus water type reveals that the highest TDS values were associated with Na-Cl, Ca-Cl and $Na-HCO_3$ water types, whereas $Ca-HCO_3$ waters had the lowest average TDS content (Fig. 4a). A comparison of methane occurrences and water types revealed that dissolved methane occurs predominantly in $Na-HCO_3$ waters for 133 samples out of 221 (Fig. 2, Fig. 4b for dissolved methane). In dissolved gas samples, the highest average methane concentrations of > 1 mM were observed in Na-Cl, $Na-HCO_3$-Cl and $Na-HCO_3$ water types, while in all other water types average methane concentrations ranged between 0.07 to 78 µM (Fig. 4c). In free gas samples, the highest average methane concentrations of >260,000 ppmv were also observed in Na-Cl, $Na-HCO_3$-Cl and $Na-HCO_3$ waters types, while in all other water types average methane concentrations ranged between 400 and 70,753 ppmv on average (Fig. 4d). The majority of free gas samples with methane concentrations > 150,000 ppmv (n = 65 out of 147) and > 0.5 mM (n=53 out of 221) in dissolved gas samples were associated with the $Na-HCO_3$ water-type. However, Fig. 4c,d consistent with Fig. 2 reveal also 5 exceptional samples with high methane concentrations in dissolved gas (n=5), free gas, or both phases (n=3) occurring in Ca-$Na-HCO_3$ (#6) and $Ca-HCO_3$ (#9) water types (see Sect. 4.4).

Ethane was also observed in some dissolved and free gas samples. In dissolved gas samples, ethane concentrations > 0.3 µM were only observed in groundwater of the Na-Cl, $Na-HCO_3$-Cl and $Na-HCO_3$ water types. Only one exception circled in Fig.

4e of a sample with elevated ethane content (0.9 µM) was observed in a Ca-$HCO_3$-Cl type sample (see Sect. 4.4). In free gas samples, ethane concentrations >100 ppmv were only observed in groundwater of the Na-Cl, Na-$HCO_3$-Cl and Na-$HCO_3$ water types with average concentrations of 79, 603, and 160 ppmv respectively (Fig. 4f). Only one exceptional sample containing ethane in free gas at 73 ppmv was found in Ca-Na-$HCO_3$ type water (Fig. 4f).
Few samples contained propane with the highest concentration of dissolved propane (0.9 µM) occurring in the Ca-$HCO_3$-Cl water type and lower propane concentrations found in the Na-Cl (0.01 µM, n= 5) > Na-$HCO_3$-Cl (0.006 µM, n=9) > Ca-$HCO_3$ (0.003, n=2) > Na-$HCO_3$ (0.002 µM, n=15) water types. For free gas samples, propane was found in Na-$HCO_3$-Cl (1.3 ppmv, n=8) > Ca-$HCO_3$ (0.6 ppmv, n=3) > Na-$HCO_3$ (0.5 ppmv, n=21) > Na-Cl (0.3 ppmv, n=4) water types.
Hence, there appears to be a correlation between the water type and the number of samples containing elevated methane and ethane concentrations in the shallow groundwaters of Alberta and a consistent relationship between gases in dissolved and free phases as shown in Humez et al. (2016).

### 3.4 Isotopic composition of groundwater

The $\delta^{18}O$ and $\delta^{2}H$ values of groundwater varied from –24.3 to –8.4 ‰ with an average of –18.5 ± 1.9 ‰ and from –190.8 to –94.2 ‰ with an average of –147.4 ± 13.1 ‰, respectively (n=222) (Fig. 5. Hydrogen and oxygen isotope values of all water samples plotted closed to the local meteoric water lines (LMWL) of Edmonton and Calgary (Peng et al., 2004) suggesting atmospheric recharge of groundwater with at most minor influence of evaporation and water-rock interactions on the isotopic composition of the groundwater.

### 3.5 Isotopic composition of dissolved constituents in groundwater

#### 3.5.1 Sulfur and Oxygen Isotope Ratios of Sulfates

The $\delta^{34}S_{SO4}$ values in groundwater ranged from –26.6 to +40.9‰ with a mean value of +1.8 ± 12.4 ‰ (n=158). The $\delta^{18}O_{SO4}$ values in groundwater ranged from –17.7 to +11.2 ‰ with a mean value of –0.6 ± 6.7 ‰ (n=138) (Table 2).

#### 3.5.2 Carbon Isotope Ratios of Dissolved Inorganic Carbon

The partial pressure of $CO_2$ ($pCO_2$) was calculated for all samples based on field pH and alkalinity based on geochemical speciation with PHREEQC (Parkurst and Appelo, 1999). $P_{CO2}$ values ranged between $10^{-8.14}$ and $10^{+0.58}$ atm and the pH values in the samples containing methane ranged from 6.5 to 10.2 with a mean value of 7.9 ± 0.8 (n=225). The $\delta^{13}C_{DIC}$ values ranged from –30.8 ‰ to elevated values of +21.2 ‰ with an average of –10.8 ± 8.7 ‰ (n=221). The highest $\delta^{13}C_{DIC}$ values of +21.2 ‰, +17.7 ‰, +15.3‰ and +14.3‰ occurred in samples with elevated methane concentrations in dissolved and free gases of >1 mM and >900,000 ppmv respectively.

### 3.5.3 Nitrogen and Oxygen Isotope Ratios of Nitrates

Only 24 samples contained sufficient nitrate for isotope analysis (Table 2). The $\delta^{15}N_{NO3}$ values varied from -10.4 to +21.8 ‰ with an average of +7.8 ± 8.2 ‰, while $\delta^{18}O_{NO3}$ values ranged from –13.2 to +25.7 ‰ with an average of –1.5 ± 12.0 ‰.

### 3.6 Isotopic composition of methane

Methane in 133 groundwater samples including replicates had a median $\delta^{13}C$ value of –66.2 ‰ with a minimum value of –92.8 ‰, a maximum of –20.5 ‰, and a mean value of –64.6 ± 14.9 ‰ in free gas phase (Table 2). The median $\delta^{13}C$ value of methane in dissolved gas samples was –65.6 ‰ with a minimum value of –85.5 ‰, a maximum of –35.8 ‰ and a mean value of –65.0 ± 10.5 ‰ (n=35) (Table 2). Fifty-eight groundwater samples had a median $\delta^2H_{CH4}$ value of –291.5 ‰ with a minimum value of –437.1 ‰ and a maximum of –80.9 ‰ in free gas phase (Table 2).

## 4 Discussion

### 4.1 Geochemical constraints of methane-containing groundwater

Figure 5 shows that aqueous geochemistry results of the groundwater samples can be explained by three end-member compositions and their respective mixtures. The first group is described by (i) groundwater samples with low chloride, sulfate, sodium, DIC concentrations and low to intermediate $\delta^{18}O_{H2O}$ and $\delta^2H_{H2O}$ values, representing a freshwater (TDS < 2000 mg/l) end-member (plotting close to the origin in. 5, named end-member #1). Samples belonging to this group had generally low methane concentrations of <0.001 mM and a wide range of nitrate concentrations. The second group represents (ii) groundwater samples with low chloride concentrations and low $\delta^{18}O_{H2O}$ and $\delta^2H_{H2O}$ values but high sodium, sulfate and DIC concentrations (blue shading in Fig. 5). Samples belonging to this group had TDS > 4000 mg/L but had mostly low methane concentrations. These samples were predominantly obtained from GOWN wells located to the east of the Edmonton-Lethbridge corridor. Only one water sample in this group had an elevated nitrate concentration of > 4 mM and two other water samples had a nitrate concentration < 0.02 mM. These three samples were taken from wells completed in the Horseshoes Canyon, Bearpaw and Belly River Formations. The samples from this group (ii) appear to be impacted by sulfide oxidation in tills and cation exchange resulting in elevated Na, DIC and sulfate concentrations (Grasby et al., 2010). The third identified group is (iii) groundwater with elevated chloride, sodium and DIC concentrations, elevated $\delta^{18}O_{H2O}$ and $\delta^2H_{H2O}$ values, and TDS values between 2000 and 4000 mg/L, but with negligible nitrate (< 0.002 mM) and sulfate concentrations (< 1 mM) (green shading in the Fig. 5). This group is composed of $Na\text{-}HCO_3$ and $Na\text{-}HCO_3\text{-}Cl$ water-types and contains the groundwater with the highest methane concentrations. The samples in this group were obtained predominantly from wells completed in coal-bearing geological formations (e.g. Belly River and Horseshoes Canyon Formations).

The groundwater compositions investigated in this study can thus be explained by mixing between the high TDS Na-$HCO_3$ and Na-$HCO_3$-Cl end-member 3 and freshwater end-member 1 and/or mixing between the freshwater end-member 1 and end-member 2 (Fig. 5).

Results presented in Figs. 2 and 4 show that water samples with elevated methane and ethane concentrations are predominantly associated with groundwaters of the Na-$HCO_3$ water-type. This is consistent with the observations made in groundwater baseline studies of Molofsky et al. (2013) in Susquehanna County (Northeastern Pennsylvania, USA) and McPhilips et al. (2014) in Chenango country (Central New York State, USA). In both studies, elevated methane concentrations (dissolved $CH_4$ > 1 mg/L or > 0.06 mM) in groundwater were predominantly found in sodium-chloride (Na-Cl) or sodium-bicarbonate (Na-$HCO_3$) water types, while calcium-bicarbonate (Ca-$HCO_3$) type groundwater had typically very low methane concentrations or no methane at all. This is consistent with the majority of the observations made in this study.

Typically, Na-dominated groundwaters are linked to two principal processes: (1) mixing with deep saline fluids (Na-Cl types) and (2) cation exchange (Na-$HCO_3$ type).

We observe that groundwater samples with elevated methane and ethane concentrations tend predominantly to end-member 3 with negligible nitrate and sulfate concentrations (green shading in Fig. 5). Therefore, the redox status of the various groundwaters was further investigated, to better constrain the conditions that may facilitate the formation and occurrence of methane and other alkane gases in shallow groundwater bodies.

**4.2 Redox-sensitive parameters and the distribution of methane in groundwater samples**

As groundwaters evolve from highly oxidized to highly reducing conditions, they undergo a sequence of redox reactions including $O_2$ consumption, denitrification, Mn- and Fe-reduction, bacterial sulfate reduction followed by methanogenesis (Appelo and Postma, 2005). Since Mn and Fe concentration data were not available in our data set, the groundwater samples collected in this study were classified into four categories depending on the concentrations of terminal electron acceptors (TEAPs), such as $O_2$ , $NO_3$, Mn, Fe, and $SO_4$, participating successively in redox reactions:

1) oxidized;
2) denitrified but still sulfate containing;
3) undergoing bacterial sulfate reduction, and;
4) methanogenic

Figure 6a shows that typically samples with elevated oxygen concentrations > 0.01 mM ($P_{O2}$ <$10^{-1.5}$ atm) do not contain methane with the exception of 16 of 221 samples. In-situ methane formation is not possible in oxygen-containing groundwater (Chapelle, 2001). Only one sample had an elevated $O_2$ content of 0.2 mM in concert with a dissolved methane concentration of 0.7 mM (Fig 6a) potentially indicating methane migration from more reducing aquifer portions (see Sect. 4.4).

Figure 6b shows that samples containing nitrate (> 0.006 mM) did not contain any dissolved methane. A significant inverse correlation was found between nitrate and methane concentrations (Kendall's *tau* = -0.167 and $p < 0.05$, Spearman's *rho* = -0.228 and $p < 0.05$, n=133) suggesting that nitrate-containing groundwaters are not suitable for methane formation or conservation. Figure 6c shows that only groundwater samples with sulfate concentrations lower than 1 mM contained elevated dissolved methane. A significant inverse correlation was found between sulfate and methane concentrations (Kendall's *tau* = -0.320 and $p < 0.05$, Spearman's *rho* = -0.463 and $p < 0.05$, n=221) suggesting that methane formation does not commence while sulfate is still present at concentration > 1 mM. An alternate explanation could be that methane migrated into the aquifers containing $O_2$, $NO_3$ and/or $SO_4$ thereby creating more reducing conditions and consuming oxygen from all these species. We suspect that noticeable amounts of methane in dissolved and free gas samples were only observed after sulfate had been removed presumably by bacterial (dissimilatory) sulfate reduction to levels < 1 mM (Fig. 6c). These observations are consistent with the redox ladder concept and are further illustrated in Fig. 7.

Figure 7 shows a cross-plot of nitrate and sulfate concentrations with concentrations of dissolved methane displayed as colored circles revealing that groundwater samples containing elevated concentrations of nitrate (>0.006 mM) and intermediate to high sulfate (1 to 1000 mM) concentrations did not contain significant amounts of dissolved methane (<0.01 mM) (light grey shading, Fig. 7). The presence of nitrate and sulfate indicates that neither complete denitrification nor complete bacterial sulfate reduction has occurred in these aquifers and hence in-situ methane formation would be in contradiction to the redox ladder concept. In this group, only four samples had non-negligible methane concentration of $0.06 < CH_4 < 1.2$ mM (circled in Fig. 7; see Sect. 4.4). This is consistent with the hypothesis that methane migrated into aquifers containing $NO_3$ and $SO_4$.

Another group of groundwater samples is characterized by elevated (> 1 mM) sulfate concentrations, low nitrate, and negligible amounts of dissolved methane (<0.01 mM) (hatched area, Fig. 7). Nearly all these samples had negligible methane concentrations except five samples (circled in Fig. 7 within the hatched area) with elevated methane concentrations > 0.01 mM. There are two possible reasons which could explain the coexistence of methane with sulfate concentrations slightly above 1 mM; either these groundwater samples never contained nitrate but the presence of sulfates indicates that bacterial sulfate reduction has not occurred yet and hence methanogenesis has not commenced; alternatively, methane may have migrated into some of these aquifers and was oxidized through denitrification explaining the lack of nitrate.

A third group of samples has negligible concentrations of nitrate (< 0.006 mM) and sulfate (< 1 mM) and contains the vast majority of samples with methane concentrations >0.1 mM including those with the highest methane content of >1.2 mM (dark grey shading, Fig. 7). This suggests that both denitrification and bacterial sulfate reduction have occurred creating redox conditions favorable for in-situ methanogenesis. These conditions were predominantly observed in Na-$HCO_3$ and Na-$HCO_3$-Cl water-types with one exception in Ca-$HCO_3$ water (circled dot in dark grey shading, Fig. 7).

### 4.3 Stable isotopes constraints

To further test the hypotheses of occurrences of redox processes such as denitrification, bacterial sulfate reduction, methanogenesis, and potentially methane oxidation, we investigated the isotopic compositions of nitrate, sulfate and DIC.

#### 4.3.1 Nitrogen and Oxygen Isotope Ratios of Nitrates

The isotopic composition of nitrate indicates predominantly the sources of this nutrient (Kendall, 1998). In one group of groundwater samples with both low $\delta^{15}N_{NO3}$ (< +10 ‰) and $\delta^{18}O_{NO3}$ values (< 0 ‰) accompanied by low nitrate concentrations (<0.06 mM) nitrate appears to be derived from nitrification of soil organic matter (Fig. 8b) (Kendall, 1998). These groundwaters are characterized by mainly low methane concentrations (<0.05 mM) except for two samples with methane concentrations of 0.2 mM and 1.3 mM. A second group of groundwater samples had low $\delta^{18}O_{NO3}$ values of nitrate (–10 ‰ < $\delta^{18}O_{NO3}$ < +5 ‰) but elevated $\delta^{15}N_{NO3}$ values (> +10 ‰, Fig. 8b). These samples were also associated with the highest nitrate concentrations of up to 21.2 mM suggesting nitrate contamination most likely from manure spreading (Rock and Mayer, 2004). These samples belong to various groundwater types and had negligible methane concentrations <0.01 mM. The third group is described by elevated $\delta^{15}N_{NO3}$ (> +5 ‰) and $\delta^{18}O_{NO3}$ values (> +20 ‰), negligible nitrate concentrations (<200 µM) and methane concentrations up to 0.3 mM. Such an isotopic signature could theoretically be sourced from $NO_3$-containing mineral fertilisers (Kendall et al., 1998) but in this case elevated nitrate concentrations would be expected. During denitrification in a closed system, it is expected that as nitrate concentrations decrease the remaining nitrate becomes progressively enriched in $^{15}N$ and $^{18}O$ (Mariotti et al., 1988; Boettcher et al., 1990). Plotting nitrate concentrations versus $\delta^{15}N$ values of nitrate containing samples provides some evidence that these samples may have been affected by denitrification (Fig. 8a). Hence, isotope analyses revealed different sources of nitrate and processes such as mixing between nitrification-derived and manure-derived end-members, but only little indication of denitrification. Only 24 of 225 samples had sufficient nitrate to conduct isotope analyses and no methane was observed in samples with elevated nitrate concentrations (Fig. 8a).

#### 4.3.2 Sulfur and Oxygen Isotope Ratios of Sulfates

During bacterial (dissimilatory) sulfate reduction (BSR) in a closed system, it is expected that sulfate concentrations decrease while $^{34}S$ and $^{18}O$ become progressively enriched in the remaining sulfate (Fritz et al., 1989). Plotting sulfate concentrations versus $\delta^{34}S_{SO4}$ values (Fig. 9a) reveals that samples with the highest sulfate concentration have $\delta^{34}S_{SO4}$ values between 0 and –10 ‰. This, together with $\delta^{18}O_{SO4}$ values < +4 ‰ suggests that these groundwater samples derive their sulfate predominantly from pyrite oxidation (Fig. 9b, Grasby et al., 2010). Many samples with lower sulfate concentrations also show $\delta^{34}S$ and $\delta^{18}O$ values of sulfate < 0 ‰ suggesting that oxidation of sulfide minerals is the sulfate source (Fig. 9b),

creating a mixing trend in Fig. 9a between a low-$SO_4$ content groundwater with a high-$SO_4$ concentration samples. There is, however, also a trend of increasing $\delta^{34}S_{SO4}$ values > +15 ‰ with decreasing sulfate concentrations (Fig. 9a) suggesting that bacterial sulfate reduction has occurred in these aquifers with the highest $\delta^{34}S$ value of +56.4 ‰ at a sulfate concentration of 0.03 mM. This is also confirmed by elevated $\delta^{18}O$ values of sulfate (Fig. 9b). The group of samples displaying evidence of BSR (Fig. 9b) contains many samples with elevated methane concentrations (>0.1 mM).

#### 4.3.3 Carbon isotope ratios of inorganic carbon

The isotopic composition of DIC is indicative of sources of carbon and processes that have generated or affected DIC (Mook, 2000). Fig. 10a reveals that the $\delta^{13}C$ values of most samples varied between –20 and –10 ‰, suggesting that the majority of the DIC is derived from a combination of oxidation of organic carbon and carbonate dissolution (Clark and Fritz, 1997). Most samples in this category have methane concentrations <0.1 mM. Fig. 10a further reveals a second group of samples with $\delta^{13}C$ values of DIC > –8 ‰ and reaching values as high as +20 ‰. The positive $\delta^{13}C$ values are clear evidence for in-situ bicarbonate-based methanogenesis within the aquifer, during which $^{12}C$ is preferentially allocated to methane while the remaining $CO_2$ and subsequently DIC becomes enriched in $^{13}C$ (Barker and Fritz, 1981). All samples in this category had methane concentrations > 1.2 mM.
Fig. 10b shows that all samples with elevated sulfate concentrations contained DIC with low $\delta^{13}C_{DIC}$ values (average –13.8 ‰). In contrast, samples with the lowest sulfate concentrations were accompanied by the highest $\delta^{13}C_{DIC}$ values of up to +21.2 ‰ and the highest methane concentrations, while samples with methane concentrations between 0.1 and 1.2 mM plot in between (Fig. 10b). This strongly supports the hypothesis that BSR needs to proceed towards completion prior to commencement of in-situ methanogenesis with the aquifer and generation of elevated methane concentrations in the aquifer.
The elevated $\delta^{13}C$ values rule out the occurrence of anaerobic methane oxidation (AOM) coupled with BSR after migration of methane into sulfate-containing aquifers (Knab *et al.*, 2009) that would result in $^{13}C$-depleted DIC, due to preferential oxidation of $^{12}CH_4$.

### 4.4 Evidence for in-situ formation, for migration, and for oxidation of methane

#### 4.4.1 Classification criteria

Using the information described above, we evaluated whether geochemical conditions in the aquifers were suitable for in-situ methane generation, or whether the geochemical conditions suggest that methane must have migrated into the aquifer. This was achieved for 135 samples that had sufficient aqueous and gas geochemistry data including the following information and parameters:

(1) Water type derived from the Piper diagram based on balanced major ion chemistry,
(2) Redox parameters such as dissolved oxygen, sulfate, and nitrate concentrations,
(3) Gas composition in dissolved and/or free gas samples,
(4) Isotope values of methane in free and/or dissolved gas samples,
(5) Geological formation in which the groundwater wells were completed.

#### 4.4.2 In-situ biogenic methane generation (category #1)

Category #1 contains samples with $\delta^{13}C_{CH4} < -55$ ‰ and a high dryness parameter >1000 indicating biogenic methane. Aqueous geochemistry data were consistent with methanogenic conditions (no nitrate, sulfate concentrations negligible) and no traces of propane were detected (Table 3). This category contains 53 of 135 samples (39 %) yielding clear evidence that biogenic methane was generated in-situ under methanogenic aquifer conditions (Fig. 11). Elevated methane and ethane concentrations where found usually in aquifers completed in the coal- and shale-bearing geological formations (e.g. Horseshoes Canyon and Belly River Group Formations). These samples are classified as $CH_4$-type A in Table 3, and Fig. 13.

The remaining 82 samples that did not fall into category #1 show at least one of the following characteristics:

1) Presence of traces of propane in dissolved and/or free gas (n = 31 for free gas, n=22 for dissolved gas);
2) Dryness parameter < 500 (n=23);
3) Elevated methane concentrations (>0.01 mM), while oxygen (>0.01 mM), sulfate (>1 mM), and/or nitrate (>0.006 mM) concentrations are not negligible (n = 6);
4) A carbon isotope ratio that may suggest thermogenic methane ($\delta^{13}C_{CH4} > -55$ ‰) (n = 24).

These characteristics may indicate that methane has migrated and potentially has undergone oxidation.

#### 4.4.3 Migration of biogenic methane into more oxidizing aquifer sections (category #2)

Category #2 contains samples with $\delta^{13}C_{CH4} < -55$ ‰, elevated dryness parameter >1000, and no traces of propane, indicating biogenic methane. However, Table 3 reveals that methane was detected in groundwater with either elevated sulfate concentrations (> 1 mM, n=3) or elevated nitrate concentrations (> 0.006 mM, n=2). This is inconsistent with conditions suitable for in-situ methanogenesis and therefore it is postulated that biogenic methane had migrated from more reducing sections of the aquifer into sections with more oxidizing conditions and has not yet been oxidized due to residence times that are short with respect to the rather slow turnover of microbial AOM (Jorgensen et al., 2001). This category contains 4 % of the samples (5 of 135 samples) and is listed as $CH_4$-type B in Table 3 and Fig. 13.

### 4.4.4 Apparent or pseudo-thermogenic methane in shallow aquifers (category #3)

Category #3 contains samples with $\delta^{13}C_{CH4} > -55$ ‰ (Fig. 11) but without detectable higher alkanes. Such high $\delta^{13}C$ values can either indicate a thermogenic gas source or may be caused by methane oxidation. An increase of $\delta^{13}C_{CH4}$ values was observed with decreasing $\delta^{13}C_{DIC}$ values (all < –5 ‰) and sulfate concentrations decreased with increasing $\delta^{34}S_{SO4}$ values (–10 ‰< $\delta^{34}S_{SO4}$ <+15 ‰) (Fig. 12). The very low methane concentrations and the absence of higher alkanes do not support a significant flow of thermogenic gas from deep geological formations below the aquifers. Instead, the data indicate that biogenic methane has been oxidized within the aquifers, possibly coupled with bacterial sulfate reduction (Fig. 12). This process enriches $^{13}C$ in the remaining methane (Baker and Fritz, 1981) imparting a $\delta^{13}C$ value that can be misinterpreted as indicating a thermogenic gas signature. The occurrence of methane oxidation is further confirmed by a cross-plot of $\delta^{13}C$ values of methane and those of $CO_2$ (Fig. 11). It is also possible that post-sampling degradation of low-methane samples occurred. Hence we conclude that all the samples in category #3, corresponding to $CH_4$-type D (Table 3), are either affected by methane oxidation or in some cases possibly by increased analytical uncertainty due to low methane concentrations. The elevated $\delta^{13}C$ values of methane are therefore not indicative of leakage of thermogenic methane from deeper portions of the stratigraphic column into shallow aquifers. Category #3 contains 13 % of the samples (17 of 135).

### 4.4.5 Thermogenic-biogenic mixed gas origin (category #4)

Category #4 contains samples with non-negligible concentrations of higher alkanes (e.g. ethane and propane) and low dryness parameter values. For all these samples, aqueous geochemistry results suggested methanogenic conditions with no nitrate and negligible sulfate concentrations. This category contains 40 % of the samples (55 of 135) and is further subdivided into 3 sub-categories (Table 3).

Sub-category 4.1 contains two samples with $\delta^{13}C_{CH4}$ values > –55 ‰ and with very low methane concentrations and traces of propane. This suggests that mixed thermogenic and biogenic gas may have migrated into overlying aquifers and may have undergone partial methane oxidation as supported by Fig. 11a where these two samples plot in the methane oxidation field. Methane in these two samples is enriched in the heavier isotopes $^{13}C$ and $^{2}H$ ($\delta^{2}H$ values > –200 ‰) supporting the methane oxidation (Coleman et al., 1981, Fig. 11b, green arrow). These samples were derived from groundwater of the Paskapoo Formation and hence are also classified as $CH_4$-type D (Table 3).

Sub-category 4.2 contains 38 samples with elevated ethane concentrations and traces of propane. All samples from subcategory #4.2 were obtained from groundwater wells completed in coal-bearing formations (e.g. Horseshoe Canyon and Belly River Formations). Five samples had $\delta^{13}C_{CH4}$ values > –55 ‰ while 33 samples had $\delta^{13}C_{CH4}$ values < –55 ‰. Cheung et al. (2010) reported that gases derived from the Horseshoe Canyon Formation in southeastern Alberta contained considerable amounts of ethane up to 4000 ppm in addition to methane with an average $\delta^{13}C$ value of –54.0 ± 4.1 ‰. This suggests that gases from the coal-bearing Horseshoe Canyon Formation contain a minor thermogenic gas component (Cheung et al., 2010). Hence the minor thermogenic gas components detected in 38 samples of this subcategory #4.2

(28.1%) appear to be mainly derived from shallow coal-bearing sedimentary units such as the Horseshoe Canyon Formation in which many of the groundwater wells are completed, and hence in-situ gas is sampled. This in-situ gas is referred to $CH_4$-type A* (Table 3, Fig. 13), since it is predominantly biogenic with only traces of thermogenic components in this mixed gas. An alternate explanation for the occurrence of ethane and propane is their microbial formation via ethanogenesis and propanogenesis (Hinrich et al., 2006). We consider this less likely since the microorganisms responsible for biological ethane and propane formations have not yet been identified.

Sub-category #4.3 contains samples with $\delta^{13}C_{CH4}$ values < –55 ‰ and a dryness parameter of < 500 or traces of propane. This subcategory contains 15 samples obtained from wells all completed in non-CBM formations. This indicates a gas that is composed of biogenic methane mixed with smaller portions of thermogenic gas was found in shallower stratigraphic units such as the Paskapoo Formation and in surficial deposits, suggesting that mixed gas has migrated upwards. These samples account for 11.1% of the investigated groundwater and are listed as $CH_4$-type C in Table 3, and Fig. 13.

No water samples in category #4 required admixture of deep (>1,000 m) thermogenic gases to explain the chemical and isotopic characteristics of dissolved and free gas samples.

#### 4.4.6 Samples with inconclusive data sets (category #5)

Five samples (4 %) could not be assigned to any of the above categories due to conflicting parameters in the aqueous or gas chemical and isotopic data sets or between replicate samples.

## Conclusion

Analysis of water types suggested that methane occurs predominantly in Na-$HCO_3$ or Na-($HCO_3$)-Cl type waters possibly indicating prolonged water-rock interaction or mixing with less mobile saline water. Taking into account the hydrochemical conditions in methane-bearing aquifers allows for a refined analysis of methane sources and a differentiation between biogenic in-situ production of methane within aquifers versus migration of biogenic or thermogenic gases into the aquifer. To achieve this we combined redox-sensitive aqueous geochemistry parameters and isotopic compositions of nitrate, sulfate and DIC with the interpretation of gas composition and isotopic fingerprints. This combined approach allowed for an improved understanding of the occurrence and distribution of methane in shallow aquifers than using carbon isotope fingerprinting and dryness parameters alone.

Low $\delta^{13}C$ values of methane combined with a high dryness parameter and methanogenic conditions indicated by aqueous geochemistry provided clear evidence for in-situ biogenic production of methane in 39% of the investigated samples ($CH_4$-type A, Fig. 13).

High dryness and biogenic C isotope signatures coexisting with elevated sulfate, nitrate and/or oxygen concentrations, and isotopic compositions of nitrate and sulfate indicating ongoing sulfate reduction and denitrification, point to a second type of

biogenic gases having formed in an anoxic milieu before migrating into more oxidizing zone aquifer section ($CH_4$-type B, 3.7% Fig. 13).

Samples with apparent thermogenic gases based on $\delta^{13}C$, $\delta^2H$ values were often characterized by no detectable higher alkanes. This suggests that these samples contained biogenic gases that had been partly oxidized, which leads to a shift towards elevated "pseudo-thermogenic" $\delta^{13}C$ values ($CH_4$-type D, 14.1 % Table 3). It is of key importance to identify the occurrence of "pseudo-thermogenic" gas signatures during monitoring of potential environmental impacts from unconventional resource development to prevent false conclusions.

For 28.1 % of the samples, ethane and sometimes propane coexisted with biogenic methane (low $\delta^{13}C$, $\delta^2H$) and these are typical for in-situ gases produced in coal seams in which the groundwater wells are completed ($CH_4$-type A*, Table 3, Fig. 13). Migration of mixed gas composed predominantly of biogenic methane with traces of propane was detected for 11.1% of the samples ($CH_4$-type C, Fig. 13) into non-CBM aquifers.

A large majority of gases (67.4 %) obtained from the GOWN network were found to be in-situ gases either derived from in-situ formation in coalbeds (28.1 %) or produced microbially within aquifers with methanogenic conditions (39.3%). We conclude that combining hydrochemistry, in particular redox-sensitive species and their isotope ratios, with gas concentration ratios and carbon isotope signatures of alkanes and $CO_2$ constitutes an excellent approach to accurately assess methane formation and migration revealing addition insights compared to approaches based on gas composition and isotope ratios only.

**Author contribution**

P. Humez and B. Mayer prepared the manuscript and implemented the geochemical and multi-isotopic approaches to determine and quantify the origin and fate of methane in the groundwater systems. M. Nightingale, A. Kingston, S. Taylor and G. Bayegnak analyzed, compiled and delivered the geochemical and isotopic databases. V. Becker, M. Nightingale, A. Kingston, S. Taylor contributed significantly to the interpretation of the geochemical and isotopic data. G. Bayegnak (AEMERA) provided expertise on the regional monitoring program while R. Millot and W. Kloppmann (BRGM) provided expertise with evaluation of geochemical processes.

**Acknowledgements**

Financial support for this project was provided by the Natural Sciences and Engineering Research Council (NSERC) of Canada, Alberta Innovates Energy & Environment Solutions (AI-EES), Alberta Environment and Parks (AEP), and the University of Calgary's Eyes High postdoctoral fellow program. We thank Steve Wallace (AEP) for his continued support and encouragement since 2006. The final year of sampling and part of the data evaluation were supported by a NSERC

strategic project grant (SPG) and by the French Agence Nationale de la Recherche (ANR) in support of the bilateral "G-Baseline" project (Environmental baseline conditions for impact assessment of unconventional gas exploitation: advancing geochemical tracer and monitoring techniques).

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

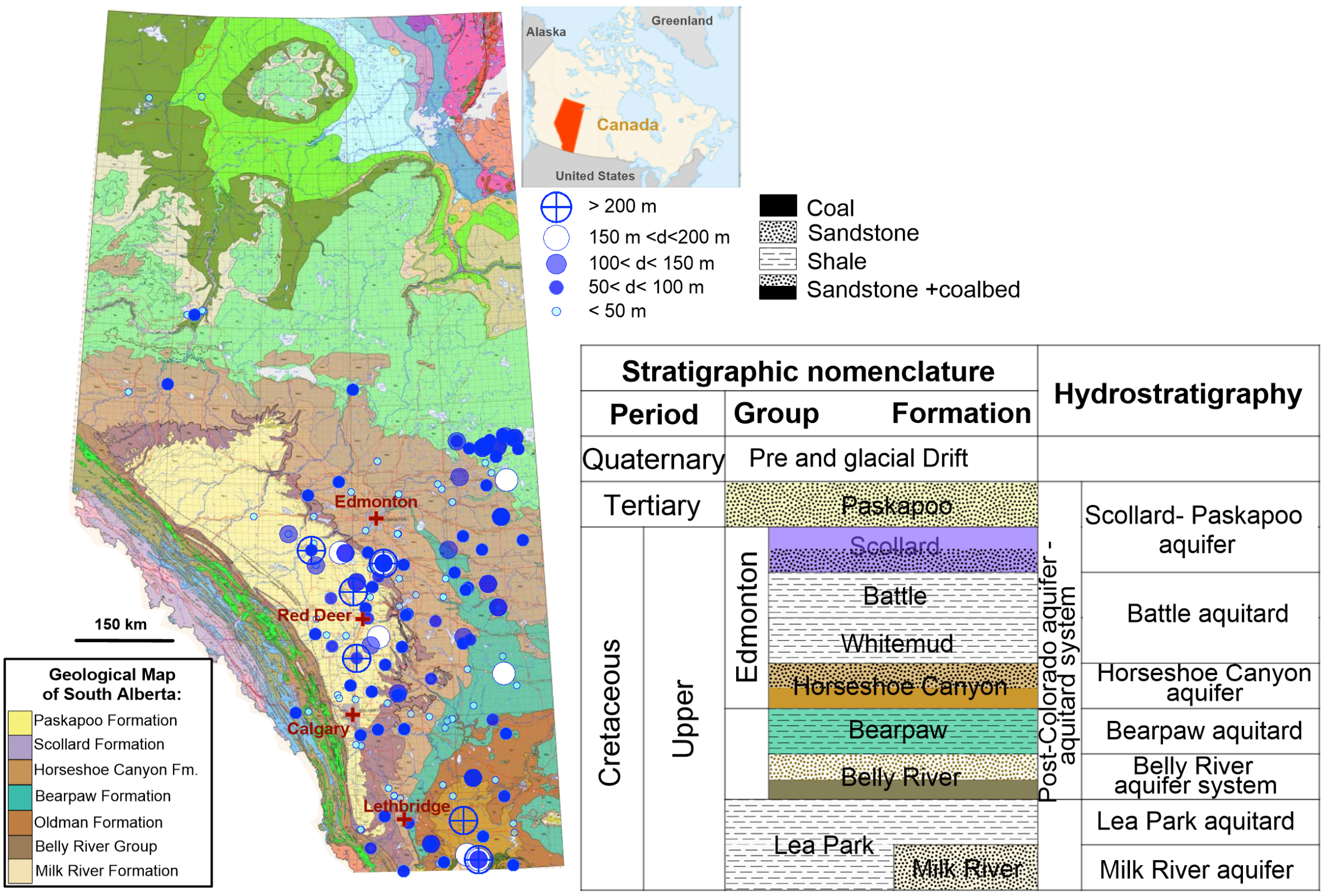

**Figure 1: Location and depths of 186 water wells from the GOWN monitoring program used in this study shown on a geological bedrock map of Alberta (Alberta Geological Survey). Also shown is a stratigraphic column for Southern Alberta (modified from Bachu, 1999).**
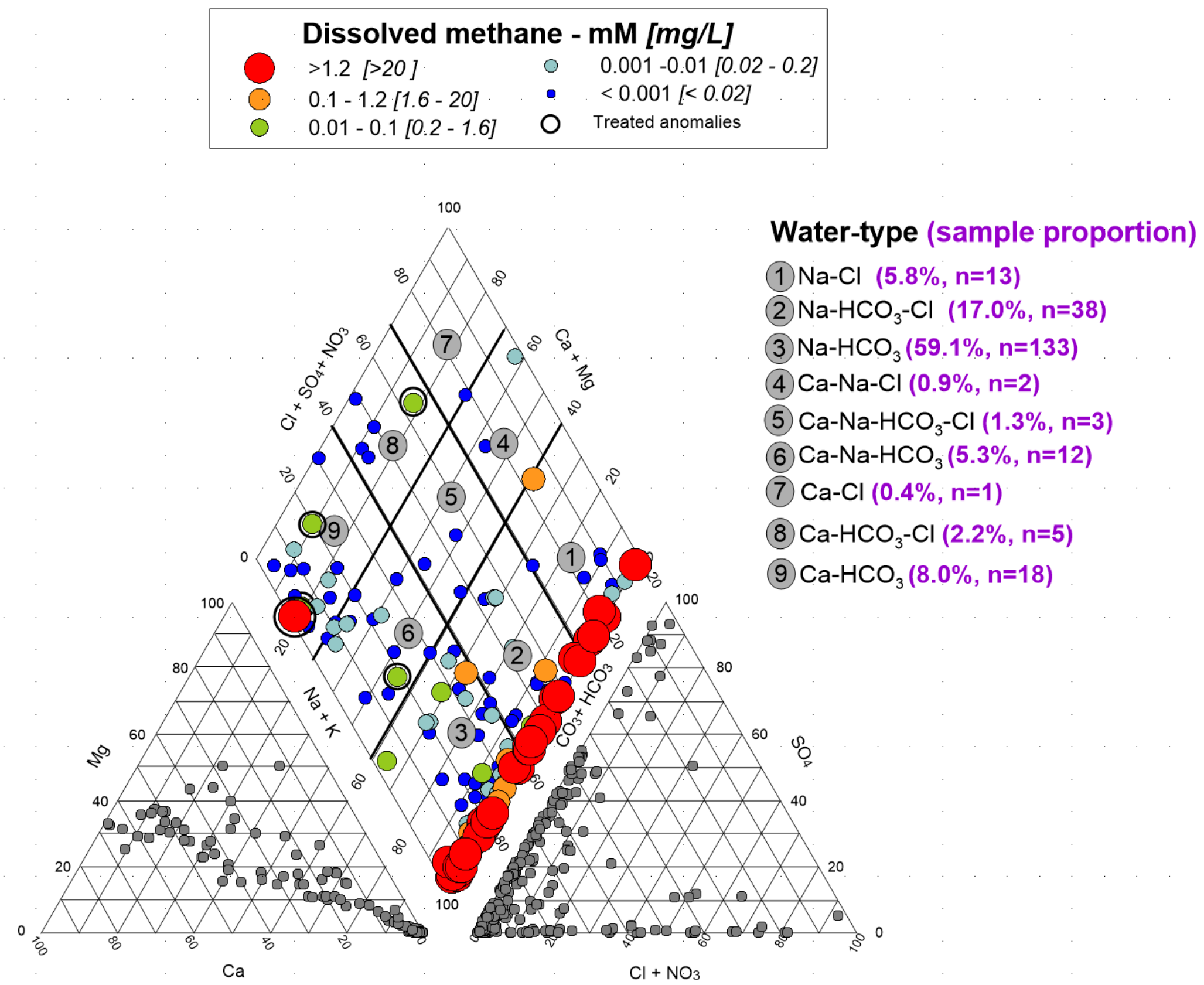

**Figure 2: Piper diagram and water type classification of groundwater samples from the GOWN network (n = 221).**

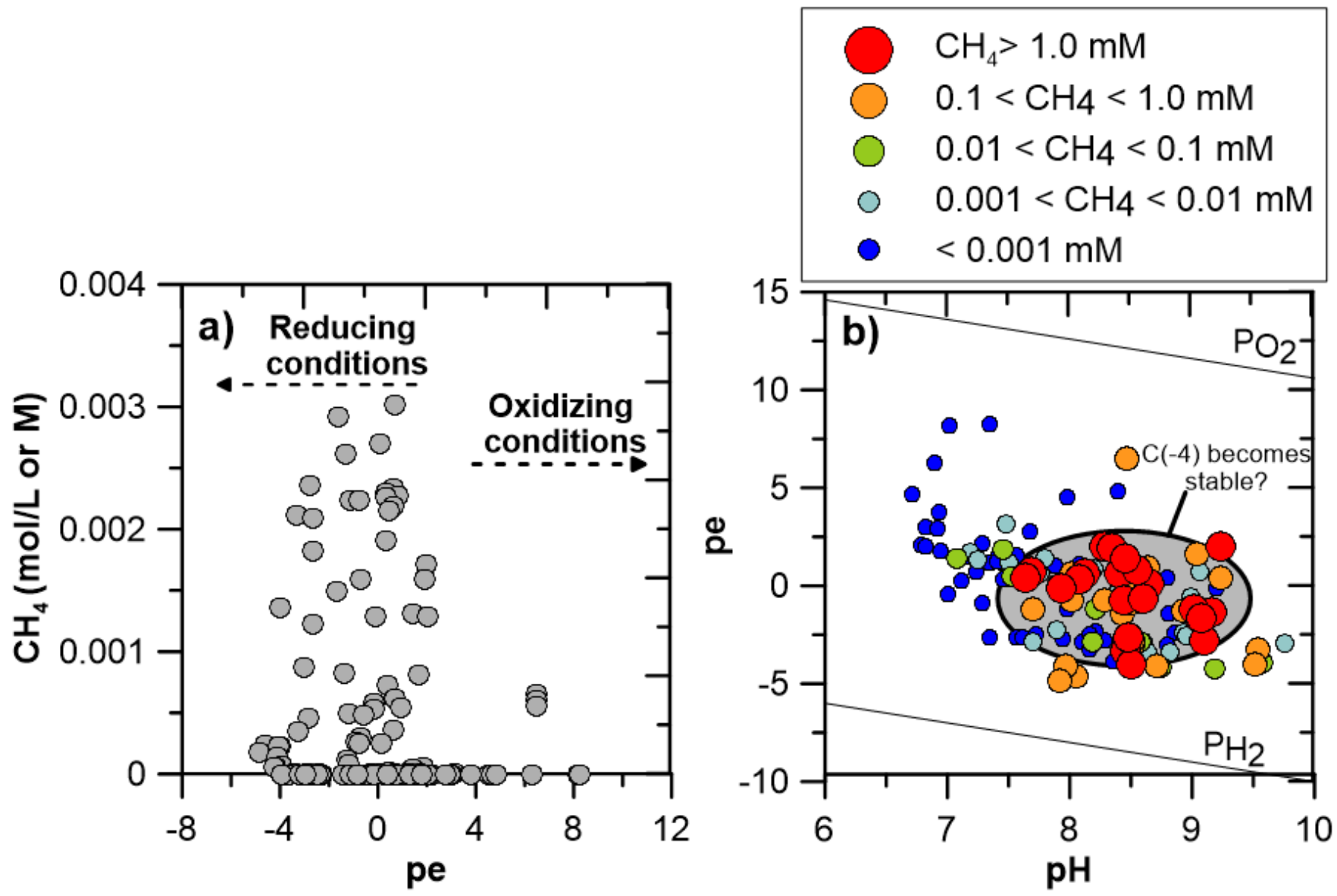

**Figure 3: a) Methane concentrations in water samples versus pe b) pe versus pH.**

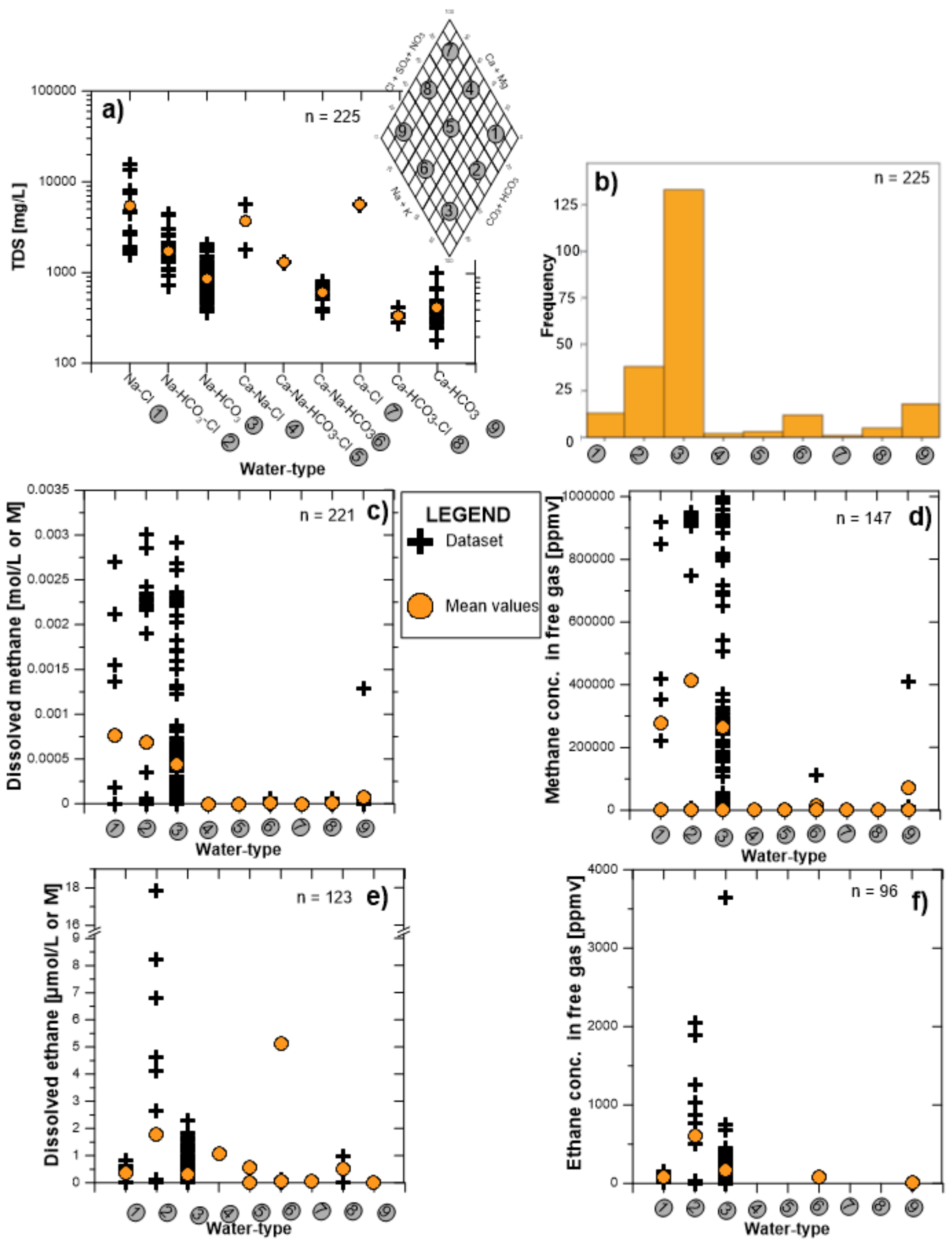

**Figure 4: a) Distribution of TDS b) frequency histogram of methane-containing samples, methane and ethane in both dissolved (c, e) and free gas (d, f) phases versus water type classification defined in the Piper diagram.**

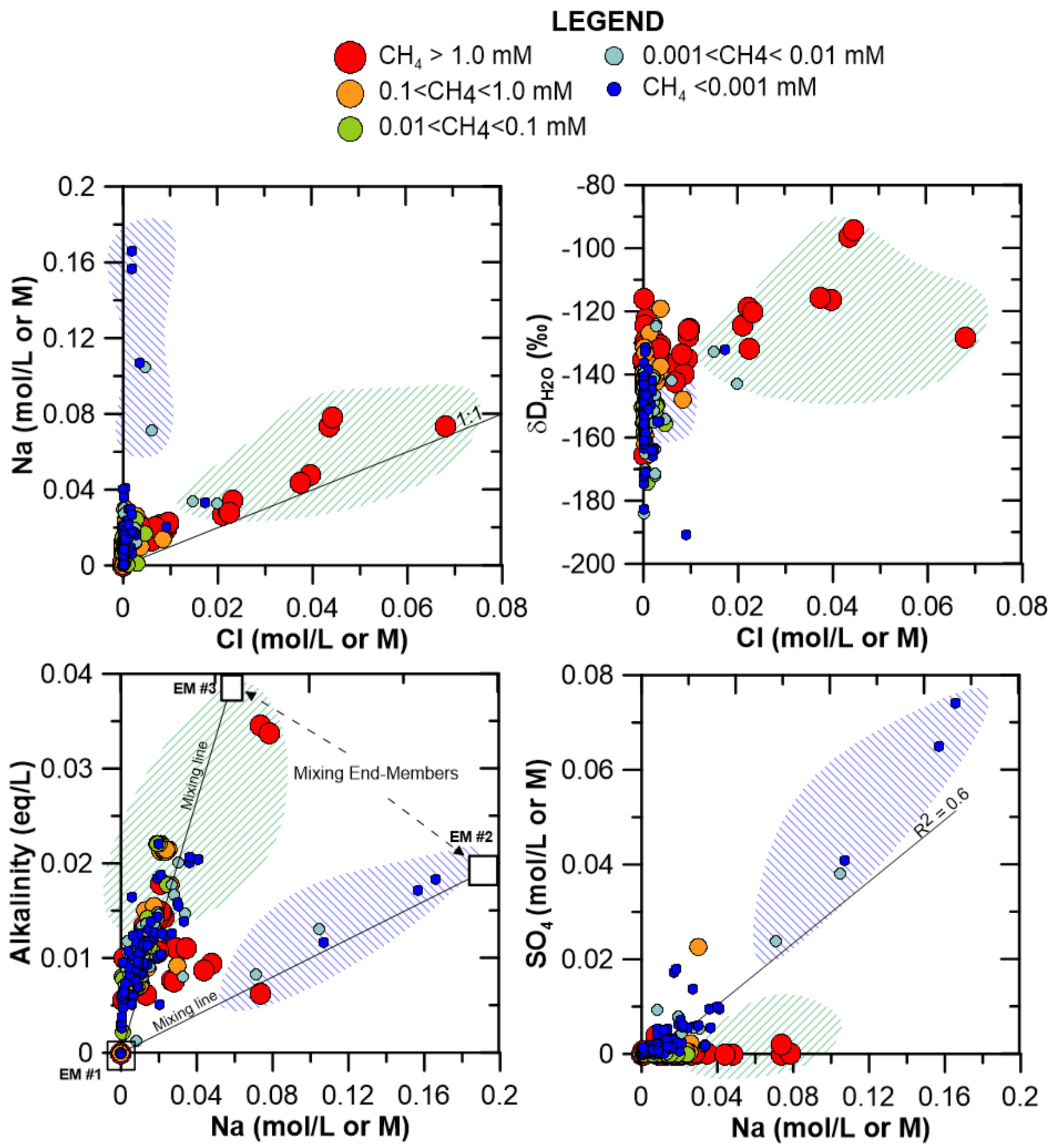

**Figure 5: Identification of groundwater geochemistry end-members mixture (shaded colors) and mixing trends indicated with solid lines. Dissolved methane concentrations are indicated by symbol size and color (EM = potential end-member compositions).**

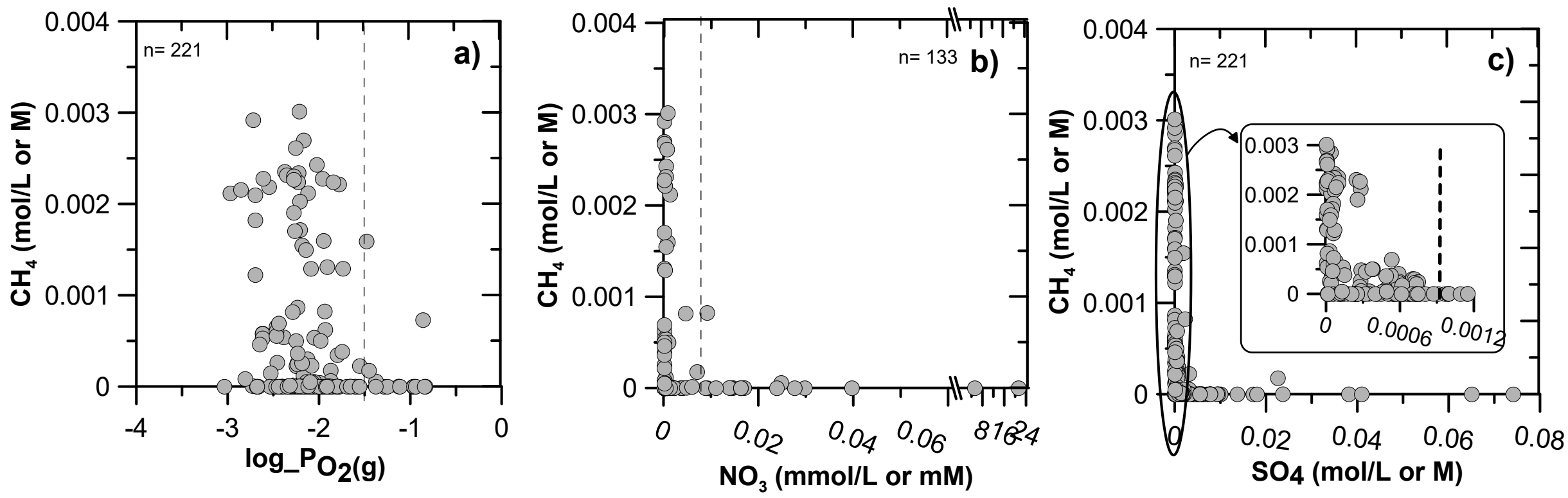

**Figure 6: Redox diagram and binary relationship between methane/nitrate and sulfate concentration in the GOWN water samples.**

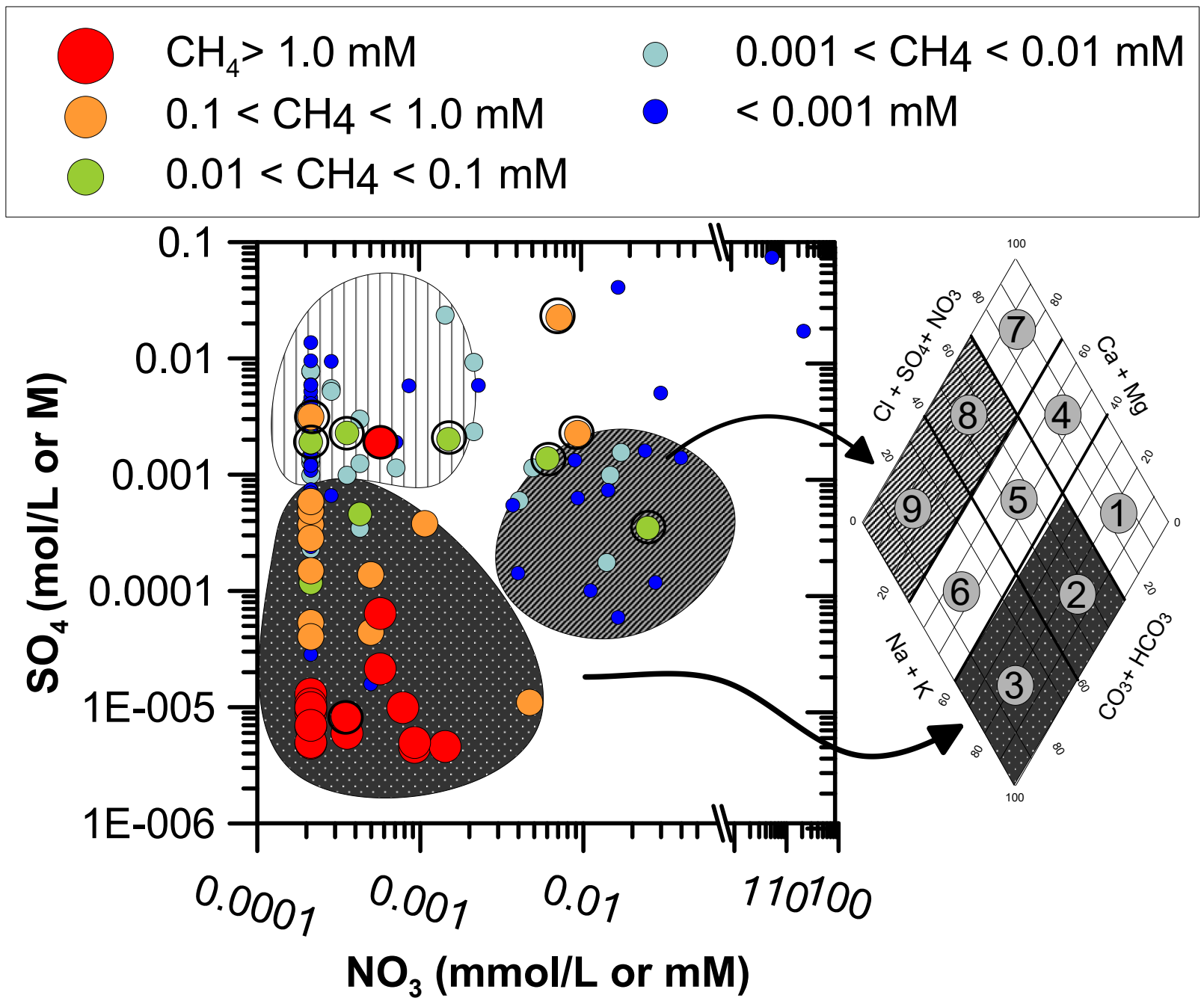

**Figure 7: Sulfate versus nitrate concentrations in water samples and methane distribution (n=133) indicated by size and color of symbols. Shaded areas relate to water types.**

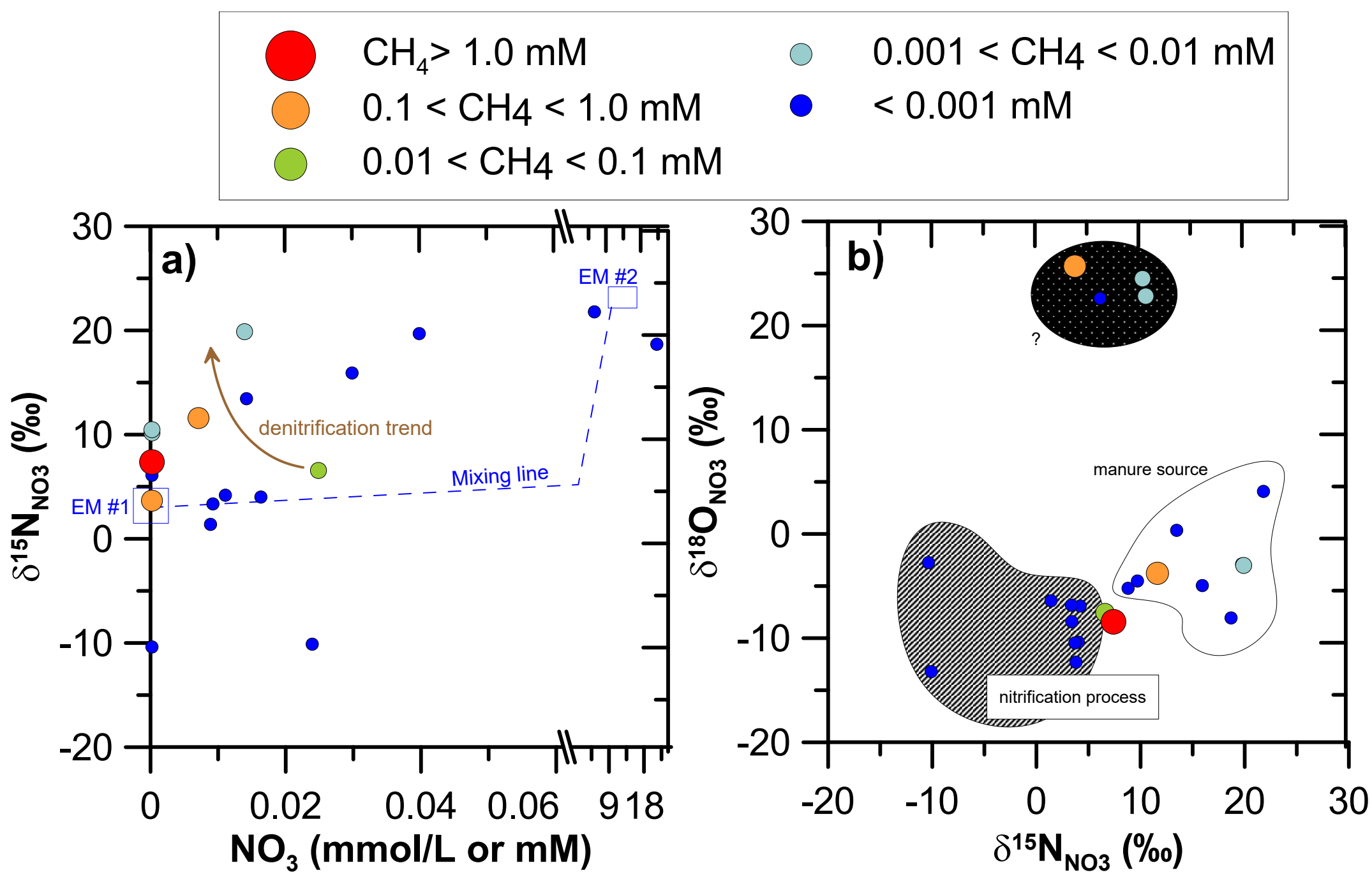

**Figure 8: Cross plot of $\delta^{15}N_{NO3}$ and $\delta^{18}O_{NO3}$ values with N-$NO_3$ concentrations (mM) and identification of three groups of data accordingly nitrate origin and formation processes. Dissolved methane concentration is reported as colored symbols.**

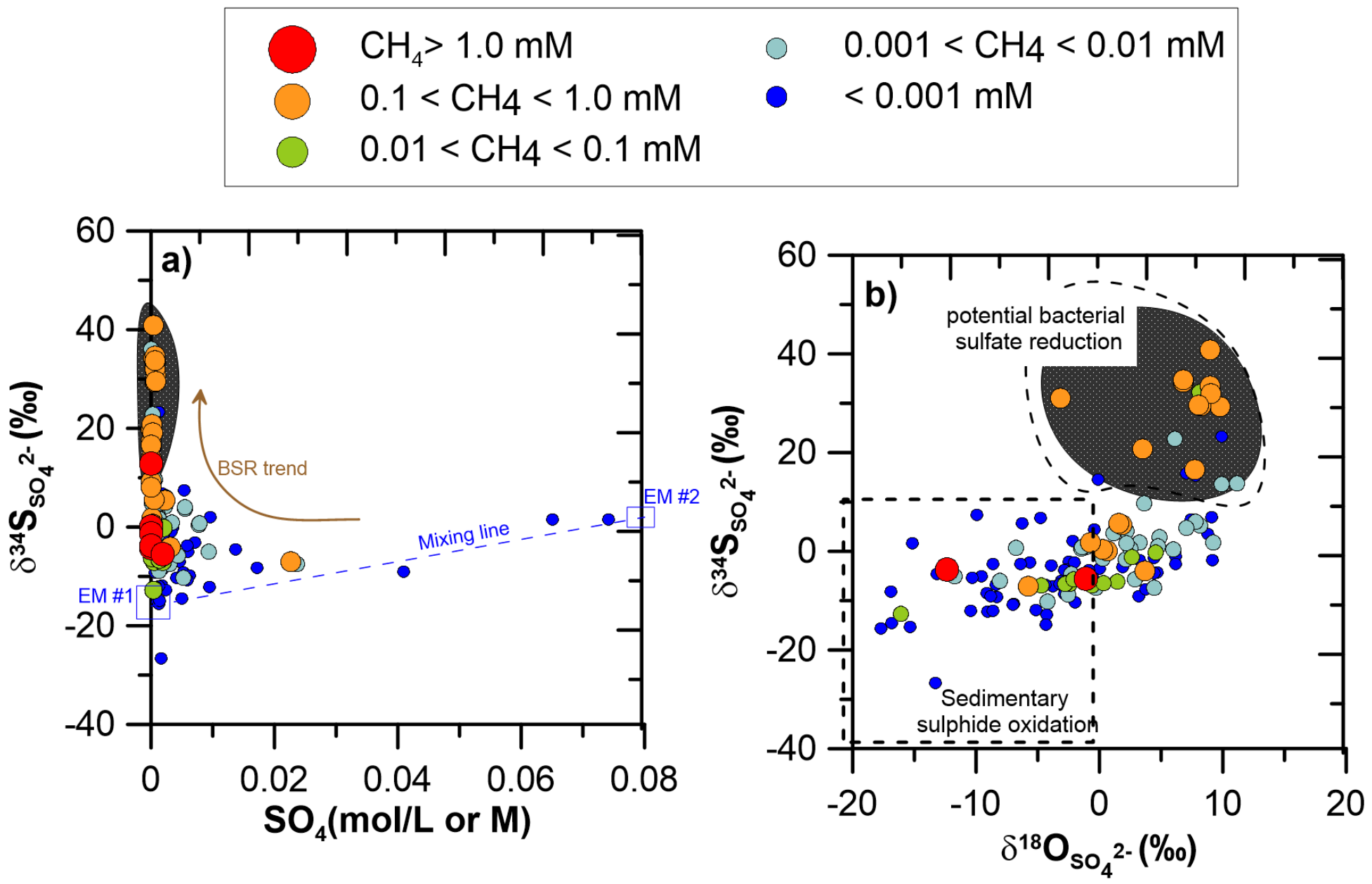

**Figure 9: Cross plot of $\delta^{34}S$ and $\delta^{18}O$ values and sulfate concentrations (M) and identification of groups of data accordingly sulfate origin and formation processes. Dissolved methane concentrations are reported as colored symbols.**

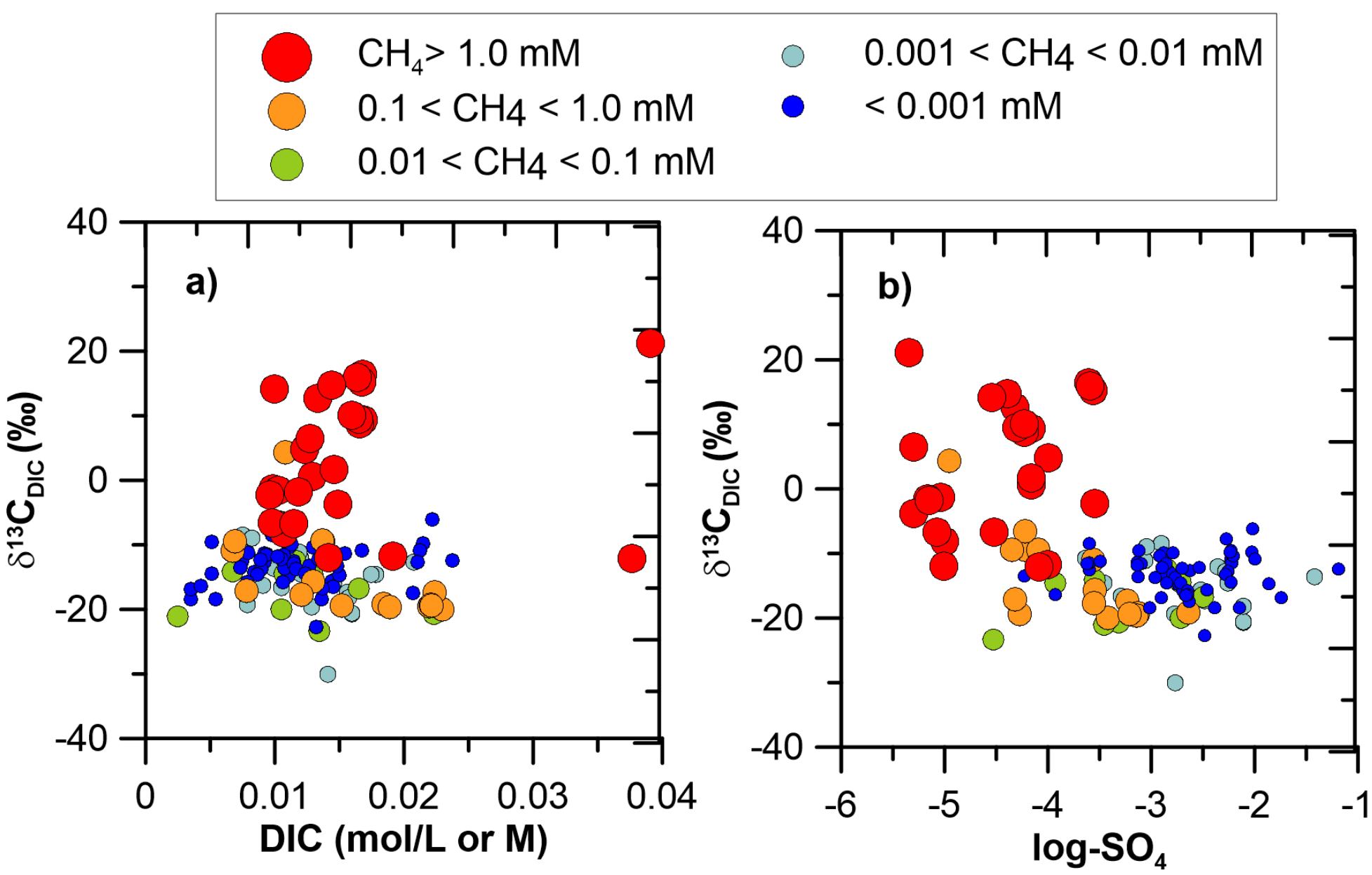

**Figure 10: δ[13]C values of DIC and methane concentrations versus a) DIC concentrations and b) sulfate concentrations (log-concentration).**

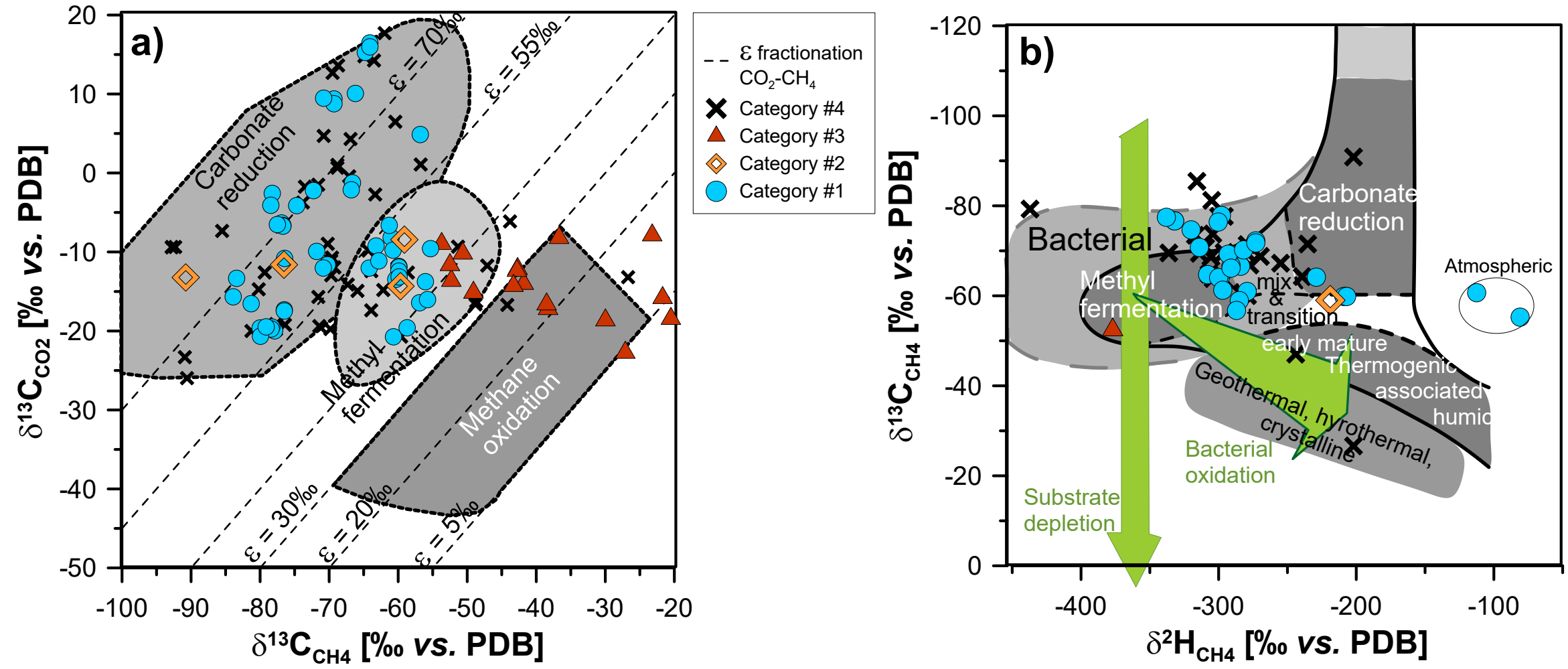

**Figure 11: a) δ[13]C values of methane and $CO_2$ revealing methane formation and consumption pathways, confirming that category #3 samples plot near the methane oxidation field, b) δ[13]C and δ[2]H values of methane for biogenic and thermogenic gas classification and their potential shift induced by secondary processes (green arrows) adapted from Whiticar (1999).**

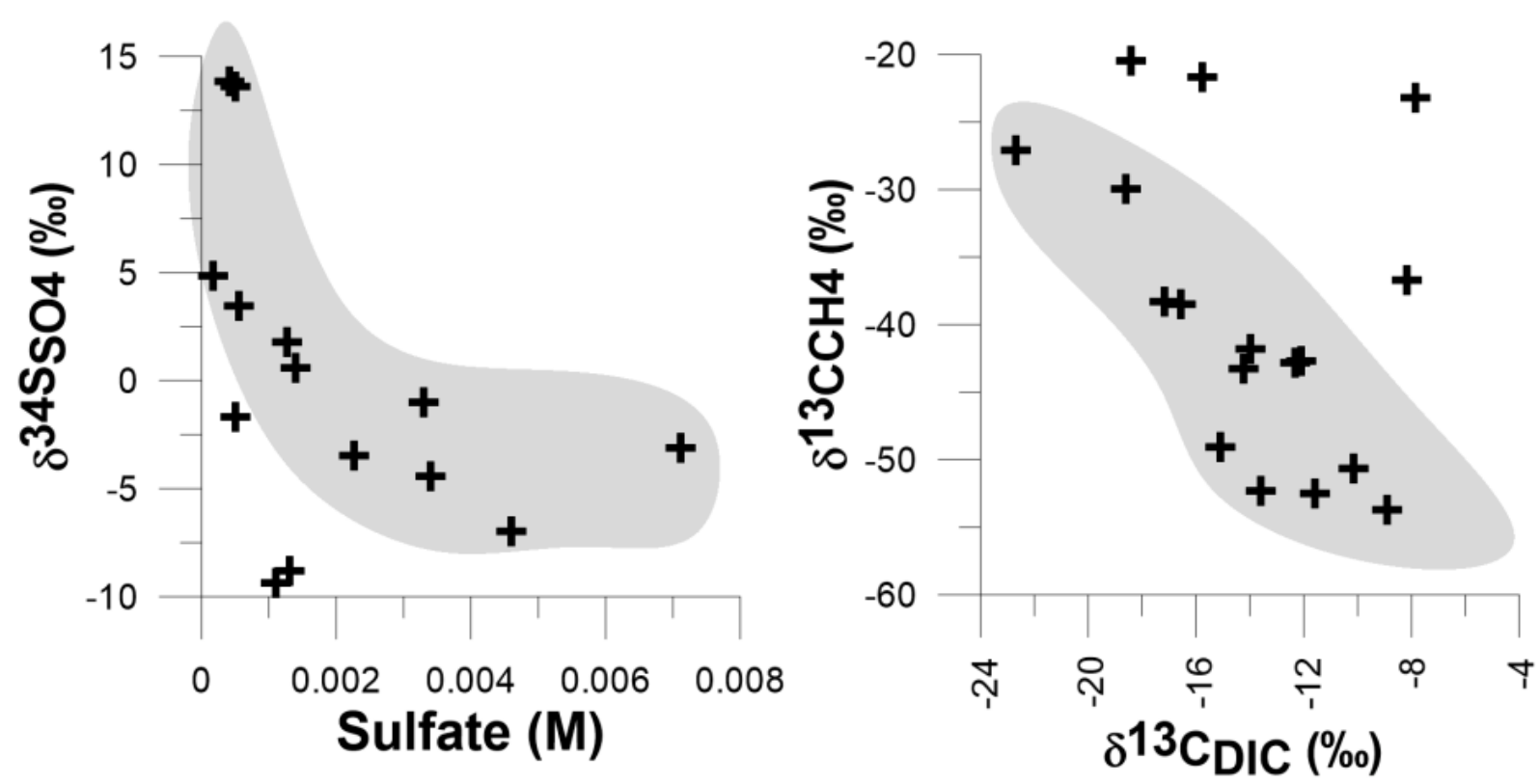

**Figure 12: δ[34]S versus sulfate concentrations and δ[13]C of methane versus DIC for samples in category #3 supporting that methane oxidation coupled with bacterial sulfate reduction is responsible for elevated δ[13]C values of methane.**

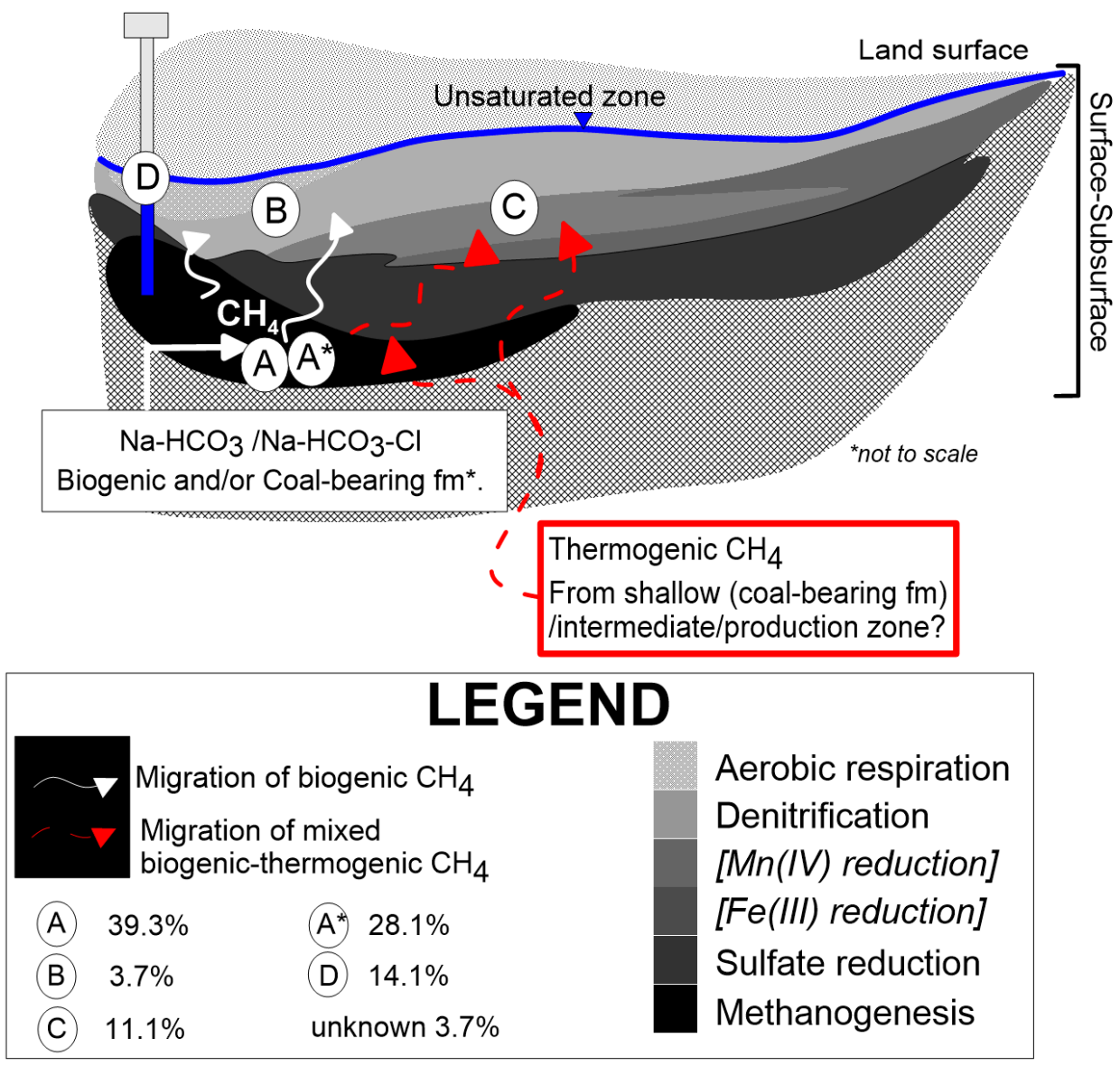

**Figure 13: Geochemical and multi-isotopic approach applied in this study to classify the methane type/occurrence considering the redox zoning constraint.**

**Table 1: Summary table of samples in the investigated data set (IB = ionic balance).**

| | Total | Criteria #1 (C#1) = -10% <IB < +10% | C#2 = C#1+ gas info | C#3= C#1 + C#2 + chemistry | |
|---|---|---|---|---|---|
| Free gas conc. only | 8 | 4 | 4 | 4 | |
| Dissolved gas conc. only | 134 | 80 | 80 | 78 | |
| Free + dissolved gas conc. | 187 | 150 | 150 | 143 | |
| No analysis available | 43*<br>50** | 8*<br>9** | / | / | |
| Isotope ratios free gas only | 141 | 105 | 105 | 100 | **135** |
| Isotope ratios dissolved gas only | 3 | 2 | 2 | 2 | |
| Isotope ratios free + dissolved gas | 36 | 34 | 34 | 33 | |
| No isotope ratio information | 192 | 101 | 93 | 90 | |
| Sum | **372** | **242** | **234** | **225** | |
| * for gas analyses<br>** for chemistry analyses | (shaded) | Retained subsets for discussion | | | |

**Table 2: Descriptive statistics for concentrations of major and minor species and for the isotopic composition of methane, nitrate, sulfate and DIC. Detection limits are also shown (DL).**

| | Unit | N | Min | Max | Range | Median | Mean | Stdev | DL |
|---|---|---|---|---|---|---|---|---|---|
| **$CH_{4,aq}$** | mM | 221 | 0.00 [2.18e-6] | 3.01 | 3.01 | 0.00 [2.77e-3] | 0.43 | 0.79 | 6.25e-7 |
| **$CH_{4,g}$** | ppmv | 147 | 0.29 | 998000 | 998000 | 38300 | 265466 | 355649 | 0.05 |
| **$C_2H_{6,aq}$** | µM | 123 | 0.00 [3.34e-4] | 17.83 | 17.83 | 0.06 | 0.60 | 1.95 | 3.33e-4 |
| **$C_2H_{6,g}$** | ppmv | 96 | 0.08 | 3650 | 3650 | 38 | 215 | 499 | 0.05 |
| **$C_3H_{8,aq}$** | µM | 32 | 0.00 [3.18e-4] | 0.90 | 0.90 | 0.00 [1.57e-3] | 0.03 | 0.16 | 2.27e-4 |
| **$C_3H_{8,g}$** | ppmv | 36 | 0.05 | 4.60 | 4.55 | 0.26 | 0.67 | 0.91 | 0.05 |
| **$O_2$** | mM | 199 | 0.01 | 0.25 | 0.25 | 0.01 | 0.02 | 0.04 | 1.56e-3 |
| **DIC** | mM | 225 | 0.81 | 39.07 | 38.25 | 12.29 | 12.94 | 5.59 | 8.20e-3 |
| **Ca** | mM | 136 | 0.01 | 9.08 | 9.06 | 0.40 | 0.94 | 1.26 | 7.49e-3 |
| **$NO_3$** | µM | 136 | 0.21 | 21290.71 | 21289.79 | 0.21 | 203.42 | 1876.17 | 0.03 |
| **K** | mM | 225 | 0.01 | 0.54 | 0.53 | 0.05 | 0.07 | 0.07 | 5.12e-3 |
| **Mg** | mM | 225 | 0.00 [2.26e-3] | 12.45 | 12.44 | 0.05 | 0.60 | 1.47 | 4.12e-6 |
| **Na** | mM | 225 | 0.03 | 165.78 | 165.75 | 12.76 | 17.28 | 20.03 | 1.30e-2 |
| **Cl** | mM | 225 | 0.01 | 68.01 | 68.00 | 0.40 | 2.81 | 7.82 | 8.46e-3 |
| **$SO_4$** | mM | 225 | 0.00 [4.55e-3] | 74.16 | 74.15 | 0.72 | 2.83 | 8.00 | 2.44e-4 |
| **$\delta^{13}C_{CH4-FG}$** | ‰ | 133 | −92.8 | −20.5 | 72.3 | −66.2 | −64.6 | 14.9 | f($CH_4$) |
| **$\delta^{2}H_{CH4-FG}$** | ‰ | 58 | −437.1 | −80.9 | 356.1 | −291.5 | −280.8 | 54.4 | f($CH_4$) |
| **$\delta^{13}C_{CH4-DG}$** | ‰ | 35 | −85.5 | −35.8 | 49.7 | −65.6 | −65.0 | 10.5 | f($CH_4$) |
| **$\delta^{15}N_{NO3}$** | ‰ | 24 | −10.4 | 21.8 | 32.2 | 7.0 | 7.8 | 8.2 | f($NO_3$) |
| **$\delta^{18}O_{NO3}$** | ‰ | 24 | −13.2 | 25.7 | 38.9 | −5.1 | −1.1 | 12.0 | f($NO_3$) |
| **$\delta^{34}S_{SO4}$** | ‰ | 158 | −26.6 | 40.9 | 67.5 | −1.6 | 1.8 | 12.6 | f($SO_4$) |
| **$\delta^{18}O_{SO4}$** | ‰ | 138 | −17.7 | 11.2 | 28.9 | −0.6 | −0.6 | 6.7 | f($SO_4$) |
| **$\delta^{13}C_{DIC}$** | ‰ | 221 | −30.8 | 21.2 | 52.0 | −12.3 | −10.8 | 8.8 | f(DIC) |
| | Range = maximum – minimum; f(X) : function of X concentration | | | | | | | | |

**Table 3: Categories classification, boundaries and methane-type (grey shading indicate anomalies, italic = boundaries, N.D non-detected, D.L detection limit)**

| **Category** | | **#1** | **#2** | **#3** | **#4** | | | **#5** |
|---|---|---|---|---|---|---|---|---|
| **Sub-cat** | | | | | **4.1** | **4.2** | **4.3** | |
| **N** | | 53 | 5 | 17 | 2 | 38 | 15 | 5 |
| **$\delta^{13}C_{CH4}$ (‰)** | *<–55* | X | X | | | X | X | X |
| | *>–55* | X | | X | X | X (n=5) | | X |
| **Methane (mM)** | *< 0.01* | X | | X (max. 5 µM) | X | X | X | X |
| | *>0.01* | X | X | | | | | |
| **Ethane (µM)** | *< 0.002* | X | X | N.D | X | X | | |
| | *>0.002* | | | | | X | X | X |
| **Propane** | *N.D* | X | X | X | | X | X | X |
| | *>D.L* | | | | X | X | X | |
| **Dryness** | | >1000 | >1000 | n.d | >1000 | <500 | <500 | |
| **$SO_4$ (mM)** | *<1* | X | | | | | | |
| | *>1* | X | X | X | X | X | X | X |
| **$NO_3$ (mM)** | *<0.006* | X | | | | | | |
| | *>0.006* | X | X | | X | X | X | X |
| **Redox ladder** | | Yes | No | Yes | Yes | Yes | Yes | Yes |
| **Geol. formation** | | various | non-CBM | various | non-CBM | CBM | non-CBM | |
| **$CH_4$-type** | | **A** | **B** | **D** | **D** | **A*** | **C** | **E** |

**A:** In-situ biogenic $CH_4$ (39.3%)
**A*:** In-situ $CH_4$ from CBM (28.1 %)
**B:** migration of biogenic $CH_4$ into more oxidizing condition (3.7%)
**C:** Mixed gas origin (11.1%)
**D:** $CH_4$ oxidation/ Post-sampling degradation of low-$CH_4$ samples (14.1%)
**E:** Unknown (3.7%)