# Peer review of "Redox controls on methane formation, migration and fate in shallow aquifers"

_Hydrology and Earth System Sciences, 2016_

## Referee Comment (RC1) · J. McIntosh (Referee) · 11 Apr 2016

Review of Humez et al. for HESS

It was a pleasure reading the manuscript by Humez et al. on coupled aqueous and gas chemistry and isotopes of groundwater in shallow aquifers in Alberta to determine processes of methane formation, removal, and migration. The study uses a multi-tracer approach that can/should be used in other study areas to identify controls on the natural and/or anthropogenic occurrence of methane in groundwater. The paper is well-written, the figures are beautiful, and the conclusions are robust. I have a few minor suggestions on improvement prior to publication, mostly grammatical errors and a few statements that need more explanation.

Specific comments:

Pg 2, Line 21, my last name is misspelled: "McInstosh et al., 2014" should be "McIntosh".

Pg 2, Line 28, our companion paper to McIntosh et al. (2014), "Hamilton et al. (2015) Hydrogeology Journal" compared water chemistry to methane concentration in groundwater in Ontario, similar to your study. Please include this reference.

Pg 3, Lines 20-23: Methanogenic systems can also have high d13C-CH4 values from closed system CO2-reduction, where most of the CO2 pool has been depleted, and d13C values of CH4 and CO2 become increasingly more positive (up to -50+ per mil in some cases). This is another way that d13C-CH4 values can seem "artificially" high, but still be methanogenic. See Bates et al. (2011) Chemical Geology. I would add a sentence on this here and point out that d13C-CO2 and/or d13C-DIC values can help distinguish these relatively positive d13C-CH4 values from methanogenic vs. thermogenic gas sources.

Pg 7, Line 28, change "adding" to "summing".

Pg 9, Line 17, "closed" should be "close" to the LMWL.

Pg 10, section 3.6. Did you measure d13C-C2 values? If so, include. This could help identify microbial oxidation of higher chain hydrocarbons, microbial production of ethane, or mixing with thermogenic gas.

Pg 11, Line 6+, we also found the highest methane values in groundwater in Ontario associated with Na-Cl and Na-HCO3 type waters (see McIntosh et al., 2014; Hamilton et al., 2015).

Figure 5: Plot endmembers on the other plots in Figure 5.

Pg 11, Line 14, Needs further explanation. Does your data (e.g. lack of Br??) differentiate between these two sources of Na-rich waters: brines versus cation exchange? Reader is left wondering which of these processes is important here, which can have implications for fluid migration vs. in-situ water-rock reactions.
Pg 12, Line 9, your results of finding high CH4 only when [SO4]<1 mM is consistent with what has been observed in deeper coalbed methane and organic-rich shale microbial gas systems. I would make this link by adding a sentence and reference to that literature. See Schlegel et al. (2011) or other reference.

Pg 13, Line 2, add "CH4" to your list of isotopic compositions investigated.

Pg 13, Section 4.3.1. The question mark for group 3 in Figure 8 needs an explanation – i.e. what does the "question mark" represent?

Pg 14, Line 20+, This part needs clarification because you go on to say, and show in Figure 11, that there are some samples with evidence of methane oxidation, whereas you say here that there is no evidence of oxidation. Be more specific here, or simply remove statement and save for later when you discuss the higher d13C-CH4 values.

Pg 15, Line 27, "has not yet been oxidized" – be more specific: e.g. there is no evidence of methane oxidation because the elevated d13C-CH4 values are not associated with low d13C-DIC values, as expected for methane oxidation.

Pg 16, Line 8, change "imparting a d13C value" to "imparting a relatively high d13C value".

Pg 16, Line 10, "It is also possible that post-sampling degradation of low-methane samples occurred." Be more specific. What could have happened (physically) and how might that have changed the isotopic values?

Pg 17, Line 23, change "interpretation of gas composition" to "interpretation of natural gas composition."

Pg 18, Line 3: add an "and" between "d13C, dD".

Pg 18, Line 7, be more specific about this statement. For example, could add text at the end of the sentence: "such as the introduction of deeply-sourced thermogenic gases into shallow aquifers."

Figure 3, part b, It's not clear what the "(C(-4)) becomes stable" label represents. Need to explain in the figure caption and/or text.

Figure 11, part b, there wasn't much (if any?) discussion of the d2H-CH4 values in the text - add. BTW - I'm no longer using this plot in my own research because I've found that it is misleading; the dD-CH4 values are low in these western, higher latitude regions not because of a shift in metabolic pathway to "methyl type fermentation", but rather because of isotopic exchange between the 2H in the CH4 and H2O (shown in several studies now). Recent microbial studies from several coalbed methane and black shale systems show that both CO2 reduction and acetoclastic methanogenesis are typically important/present. I have a paper in review on this topic that will hopefully be published soon. In the meantime, see Bates et al. (2011) Chemical Geology for a reference.

Figure 12, part b, need to subscript "CH4" in the y-axis label.

---

## Referee Comment (RC2) · Anonymous Referee #2 · 4 May 2016

I have reviewed this manuscript quite thoroughly and find that it is in very good shape for publication as it stands. It is clean and free of typographical errors, and the figures are in very good shape. I have only one suggestion for Figures 6, 7 and 9, where SO4 and DIC are given in mol/L, that they be converted to mmol/L to be consistent with other parameters and with discussion in the text. Figure 12 could use some refinement of the ordinate axis labels. Table 2 requires some attention to significant figures, particularly for nitrate, which is reported for some data to 7 sig figs., and methane with 6 for some. I have not marked up the PDF, as I had no corrections to annotate.

Overall, this is a very nice summary of an extensive data set. They have taken steps to remove unreliable data and used appropriate statistical techniques to analyze the data. This will be a valuable contribution to approaches to understand the origin of methane in shallow groundwaters.

---

## Author Comment (AC1) · 19 May 2016

**Redox controls on methane formation, migration and fate in shallow aquifers" by P. Humez et al.**

**- Answer to comments by reviewer 1 (Dr. McIntosh) -**

**Dear Dr. McIntosh:**

**Thank you for having reviewed this manuscript and for your very constructive comments. Please find below our responses for each of your specific comments indicated as AC (Authors Comments) in bold.**

Review of Humez et al. for HESS

It was a pleasure reading the manuscript by Humez et al. on coupled aqueous and gas chemistry and isotopes of groundwater in shallow aquifers in Alberta to determine processes of methane formation, removal, and migration. The study uses a multitracer approach that can/should be used in other study areas to identify controls on the natural and/or anthropogenic occurrence of methane in groundwater. The paper is well-written, the figures are beautiful, and the conclusions are robust. I have a few minor suggestions on improvement prior to publication, mostly grammatical errors and a few statements that need more explanation.

Specific comments

Pg 2, Line 21, my last name is misspelled: "McInstosh et al., 2014" should be "McIntosh".
**AC: This correction has been made in the revised version of our manuscript.**

Pg 2, Line 28, our companion paper to McIntosh et al. (2014), "Hamilton et al. (2015) Hydrogeology Journal" compared water chemistry to methane concentration in groundwater in Ontario, similar to your study. Please include this reference.
**AC: This reference has been added to the revised version of our manuscript.**

Pg 3, Lines 20-23: Methanogenic systems can also have high d13C-CH4 values from closed system CO2-reduction, where most of the CO2 pool has been depleted, and d13C values of CH4 and CO2 become increasingly more positive (up to -50+ per mil in some cases). This is another way that d13C-CH4 values can seem "artificially" high, but still be methanogenic. See Bates et al. (2011) Chemical Geology. I would add a sentence on this here and point out that d13C-CO2 and/or d13C-DIC values can help distinguish these relatively positive d13C-CH4 values from methanogenic vs. thermogenic gas sources.
**AC: This is a very good point. To address this we have added the following statements to the revised version of our manuscript: "Methanogenic systems can also be characterized by high $\delta^{13}C_{CH4}$ values and thus by a pseudo-thermogenic methane isotope signature as a results of $CO_2$ reduction in a closed system. Under such circumstances, $\delta^{13}C$ values of $CH_4$ and $CO_2$ both increase as the $CO_2$ pool becomes progressively depleted (Whiticar et al., 1986; Whiticar, 1999; Bates et al., 2011). Hence, $\delta^{13}C_{CO2}$ and $\delta^{13}C_{DIC}$ values constitute an additional parameter that can help to distinguish whether elevated $\delta^{13}C$ values of methane are associated with biogenic methane formation or thermogenic gas sources (Whiticar et al., 1986; Whiticar, 1999)".**

Pg 7, Line 28, change "adding" to "summing".

**AC: This correction has been made in the revised version of our manuscript.**

Pg 9, Line 17, "closed" should be "close" to the LMWL.
**AC: This correction has been made in the revised version of our manuscript.**

Pg 10, section 3.6. Did you measure d13C-C2 values? If so, include. This could help identify microbial oxidation of higher chain hydrocarbons, microbial production of ethane, or mixing with thermogenic gas.
**AC: This is a very good point. The C isotope ratios of ethane have been measured on 19 samples. The plot of concentrations versus $\delta^{13}$C values of ethane is displayed below. For a sub-set of samples there is a trend of increasing $\delta^{13}$C values with decreasing ethane concentration, consistent with ethane oxidation. All these samples are from the "mixed category", and we feel that this observation does not add much new insight into the existing discussion. Hence, we prefer not to add this information unless the editor and/or reviewer strongly suggest otherwise.**

[Figure]

Pg 11, Line 6+, we also found the highest methane values in groundwater in Ontario associated with Na-Cl and Na-HCO3 type waters (see McIntosh et al., 2014; Hamilton et al., 2015).
**AC: Thank you for pointing this out. We have added the above references to this statement.**

Figure 5: Plot end-members on the other plots in Figure 5.
**AC: The end-members have been integrated in the new version of Figure 5.**

Pg 11, Line 14, Needs further explanation. Does your data (e.g. lack of Br??) differentiate between these two sources of Na-rich waters: brines versus cation exchange? Reader is left wondering which of these processes is important here, which can have implications for fluid migration vs. in-situ water-rock reactions.
**AC: The Br concentrations have been analyzed and we have created a plot of Cl/Br ratios vs Cl concentrations (see plot below) with current seawater shown as a dashed blue line.**

The Cl/Br ratios of our samples are in the range of ratios previously reported for other groundwater samples from Canada and United States (blue area in Figure below according to Gue et al. 2015; Davis et al., 2004). We stated in the original manuscript that Na-rich waters are due to either mixing with deep saline water and/or cation exchange. The plot below provides further initial insights by suggesting that samples with elevated Na and Cl concentrations appear to be mixtures between low TDS groundwater and high TDS formation waters (see crosses in Figure below, Connelly et al. 1990). We also observed that 7 of 8 samples with Cl concentration > 200 mg/L fall in the highest methane concentration category of > 1.0 mM and belong to category #4. This provides an initial indication that in select cases elevated methane in groundwater may be associated with admixture of deeper saline water rather than associated with Na-bicarbonate water type due to cation exchange processes. However, this hypothesis about methane migration pathways requires significant further testing based on regional hydrodynamic and geochemical data, which we intend to conduct in 2016. At this point, we feel that we have insufficient clear evidence to further discuss these options, and therefore intend to delete the sentences "……".

[Figure]

Pg 12, Line 9, your results of finding high CH4 only when [SO4]<1 mM is consistent with what has been observed in deeper coalbed methane and organic-rich shale microbial gas systems. I would make this link by adding a sentence and reference to that literature. See Schlegel et al. (2011) or other reference.

AC: Thank you for this valuable suggestion. Following your suggestion we have added the following statement and reference to the revised version of our manuscript: "This result is

consistent with what has been observed in deeper coalbed methane and organic-rich shale microbial gas systems (Schlegel et al., 2011)."

Pg 13, Line 2, add "CH4" to your list of isotopic compositions investigated.
**AC: This has been changed in the revised version of our manuscript.**

Pg 13, Section 4.3.1. The question mark for group 3 in Figure 8 needs an explanation– i.e. what does the "question mark" represent?
**AC: This question mark has been removed from Fig. 8 in the revised version of our manuscript to avoid confusion.**

Pg 14, Line 20+, This part needs clarification because you go on to say, and show in Figure 11, that there are some samples with evidence of methane oxidation, whereas you say here that there is no evidence of oxidation. Be more specific here, or simply remove statement and save for later when you discuss the higher d13C-CH4 values.
**AC: This statement has been deleted here and has been moved into section 4.4.4.4 of the revised version of our manuscript.**

Pg 15, Line 27, "has not yet been oxidized" – be more specific: e.g. there is no evidence of methane oxidation because the elevated d13C-CH4 values are not associated with low d13C-DIC values, as expected for methane oxidation.
**AC: As suggested, this sentence has been added in the revised version of our manuscript.**

Pg 16, Line 8, change "imparting a d13C value" to "imparting a relatively high d13C value".
**AC: This modification has been integrated in the revised version of our manuscript.**

Pg 16, Line 10, "It is also possible that post-sampling degradation of low-methane samples occurred." Be more specific. What could have happened (physically) and how might that have changed the isotopic values?
**AC: The statement "post-sampling degradation of low-methane samples occurred" refers to potential slow diffusive gas loss from sampling containers resulting in $^{13}$C enrichment in the residual methane. This has been clarified in the revised version of the manuscript.**

Pg 17, Line 23, change "interpretation of gas composition" to "interpretation of natural gas composition."
**AC: This correction has been added to the revised version of our manuscript.**

Pg 18, Line 3: add an "and" between "d13C, dD".
**AC: This has been corrected in the revised version of our manuscript.**

Pg 18, Line 7, be more specific about this statement. For example, could add text at the end of the sentence: "such as the introduction of deeply-sourced thermogenic gases into shallow aquifers."
**AC: We have followed the advice of the reviewer and have added this statement to the revised version of our manuscript.**

Figure 3, part b, It's not clear what the "(C(-4)) becomes stable" label represents. Needto explain in the figure caption and/or text.
**AC: This note has been removed from Fig. 3 in the revised version of our manuscript to avoid confusion.**

Figure 11, part b, there wasn't much (if any?) discussion of the d2H-CH4 values in the text - add.

BTW - I'm no longer using this plot in my own research because I've found that it is misleading; the dD-CH4 values are low in these western, higher latitude regions not because of a shift in metabolic pathway to "methyl type fermentation", but rather because of isotopic exchange between the 2H in the CH4 and H2O (shown in several studies now). Recent microbial studies from several coalbed methane and black shale systems show that both CO2 reduction and acetoclastic methanogenesis are typically important/present. I have a paper in review on this topic that will hopefully be published soon. In the meantime, see Bates et al. (2011) Chemical Geology for a reference.

**AC: We fully agree with the reviewer's reasoning especially since the $\delta^2H$ values of groundwater in Alberta are very negative. We therefore have removed Figure 11b and the associated discussion in the text.**

Figure 12, part b, need to subscript "CH4" in the y-axis label.

**AC: This subscript has been added in the y-axis label of Fig. 12 in the revised version of our manuscript.**

**We hope that these changes address your concerns in a satisfactory manner.**

**With best regards on behalf of all co-authors,**

**Dr. Pauline Humez**

---

## Author Comment (AC2) · 19 May 2016

Dear reviewer:

Thank you for your positive feedback and your constructive comments. We have addressed your specific suggestions in the revised manuscript as follows:

- We have changed the units in Figures 6, 7 and 9 to mmol/L as suggested;

- The ordinate label for Figure 12 has been refined by using mmol/L as a unit for sulfate concentration and by changing the number increments on the x-axis of both graphs ($\Delta x = 1$ mmol/L for the sulfate concentration axis and $\Delta x = 2$ ‰ for $\delta 13 CDIC$ axis).

- Table 2 has been changed to show significant digits consistent with the stated measurement uncertainty of the respective parameters;

[Figure]

We have also made additional minor changes in other figures to ensure consistency with respect to units as used throughout the text.

We hope that these changes address your concerns in a satisfactory manner.

With best regards on behalf of all co-authors,

Dr. Pauline Humez

---

## Author Response (AR1)

**Redox controls on methane formation, migration and fate in shallow aquifers" by Humez et al.**

**Point-by-point reply to the comments**

**Dear Editor,**

**Thank you for having accepted our manuscript submitted to HESS with minor revisions. Please find below our point-by-point reply to the comments by the reviewers with the location of modifications (line and page) as they appear in the revised manuscript.**

**We hope that these changes address the concerns of the editor and the reviewers in a satisfactory manner.**

**With best regards on behalf of all co-authors,**

**Dr. Pauline Humez**

**From reviewer #1 (Dr. McIntosh)**

| # | Comments/**Authors Comments (AC)** | Revised manuscript location |
|---|---|---|
| 1 | Pg 2, Line 21, my last name is misspelled: "McInstosh et al., 2014" should be "McIntosh". | |
| | **AC: This correction has been made in the revised version of our manuscript.** | Line 21 p 2 |
| 2 | Pg 2, Line 28, our companion paper to McIntosh et al. (2014), "Hamilton et al. (2015) Hydrogeology Journal" compared water chemistry to methane concentration in groundwater in Ontario, similar to your study. Please include this reference. | |
| | **AC: This reference has been added to the revised version of our manuscript.** | Line 22 p 2 |
| 3 | Pg 3, Lines 20-23: Methanogenic systems can also have high d13C-CH4 values from closed system CO2-reduction, where most of the CO2 pool has been depleted, and d13C values of CH4 and CO2 become increasingly more positive (up to -50+ per mil in some cases). This is another way that d13C-CH4 values can seem "artificially" high, but still be methanogenic. See Bates et al. (2011) Chemical Geology. I would add a sentence on this here and point out that d13C-CO2 and/or d13C-DIC values can help distinguish these relatively positive d13C-CH4 values from methanogenic vs. thermogenic gas sources. | |
| | **AC: This is a very good point. To address this we have added the following statements to the revised version of our manuscript: "Methanogenic systems can also be characterized by high $\delta^{13}C_{CH4}$ values and thus by a pseudo-thermogenic methane isotope signature as a results of $CO_2$ reduction in a closed system. Under such circumstances, $\delta^{13}C$ values of $CH_4$ and $CO_2$ both increase as the $CO_2$ pool becomes progressively depleted (Whiticar et al., 1986; Whiticar, 1999; Bates et al., 2011). Hence, $\delta^{13}C_{CO2}$ and $\delta^{13}C_{DIC}$ values constitute an additional parameter that can help to distinguish whether elevated $\delta^{13}C$ values of methane are associated** | Line 21-29 p 3 |

| | | |
|---|---|---|
| | with biogenic methane formation or thermogenic gas sources (Whiticar et al., 1986; Whiticar, 1999)”. | |
| 4 | Pg 7, Line 28, change "adding" to "summing". | |
| | **AC: This correction has been made in the revised version of our manuscript.** | Line 2 p 8 |
| 5 | Pg 9, Line 17, "closed" should be "close" to the LMWL. | |
| | **AC: This correction has been made in the revised version of our manuscript.** | Line 23 p 9 |
| 6 | Pg 10, section 3.6. Did you measure d13C-C2 values? If so, include. This could help identify microbial oxidation of higher chain hydrocarbons, microbial production of ethane, or mixing with thermogenic gas. | |
| | **AC: This is a very good point. The C isotope ratios of ethane have been measured on 19 samples. The plot of concentrations versus $\delta^{13}C$ values of ethane is displayed below. For a sub-set of samples there is a trend of increasing $\delta^{13}C$ values with decreasing ethane concentration, consistent with ethane oxidation. All these samples are from the "mixed category", and we feel that this observation does not add much new insight into the existing discussion. Hence, we prefer not to add this information unless the editor and/or reviewer strongly suggest otherwise.** | No changes have been made in the revised manuscript |
| |  | |
| 7 | Pg 11, Line 6+, we also found the highest methane values in groundwater in Ontario associated with Na-Cl and Na-HCO3 type waters (see McIntosh et al., 2014; Hamilton et al., 2015). | |
| | **AC: Thank you for pointing this out. We have added the above references to this statement.** | Line 15 p 11 |
| 8 | Figure 5: Plot end-members on the other plots in Figure 5. | |
| | **AC: The end-members have been integrated in the new version of Figure 5.** | See revised Fig. 5 |
| 9 | Pg 11, Line 14, Needs further explanation. Does your data (e.g. lack of Br??) differentiate between these two sources of Na-rich waters: brines versus cation exchange? Reader is left wondering which of these processes is important here, | |

| | | |
|---|---|---|
| which can have implications for fluid migration vs. in-situ water-rock reactions. | |
| **AC: The Br concentrations have been analyzed and we have created a plot of Cl/Br ratios vs Cl concentrations (see plot below) with current seawater shown as a dashed blue line. The Cl/Br ratios of our samples are in the range of ratios previously reported for other groundwater samples from Canada and United States (blue area in Figure below according to Gue et al. 2015; Davis et al., 2004). We stated in the original manuscript that Na-rich waters are due to either mixing with deep saline water and/or cation exchange. The plot below provides further initial insights by suggesting that samples with elevated Na and Cl concentrations appear to be mixtures between low TDS groundwater and high TDS formation waters (see crosses in Figure below, Connelly et al. 1990). We also observed that 7 of 8 samples with Cl concentration > 200 mg/L fall in the highest methane concentration category of > 1.0 mM and belong to category #4. This provides an initial indication that in select cases elevated methane in groundwater may be associated with admixture of deeper saline water rather than associated with Na-bicarbonate water type due to cation exchange processes. However, this hypothesis about methane migration pathways requires significant further testing based on regional hydrodynamic and geochemical data, which we intend to conduct in 2016. At this point, we feel that we have insufficient clear evidence to further discuss these options, and therefore have delete the sentence "Typically, Na-dominated groundwaters are linked to two principal processes: (1) mixing with deep saline fluids (Na-Cl types) and (2) cation exchange (Na-HCO₃ type). ".** | Some text was deleted to simply this discussion (line 21, p 11) |

[Figure]

| | | |
|---|---|---|
| 10 | Pg 12, Line 9, your results of finding high CH4 only when [SO4]<1 mM is consistent with what has been observed in deeper coalbed methane and organic-rich shale microbial gas systems. I would make this link by adding a sentence and reference to that literature. See Schlegel et al. (2011) or other reference. | |
| | **AC: Thank you for this valuable suggestion. Following your suggestion we have added the following statement and reference to the revised version of our manuscript: "This result is consistent with what has been observed in deeper coalbed methane and organic-rich shale microbial gas systems (Schlegel et al., 2011)."** | Line 16-17 p 12 |
| 11 | Pg 13, Line 2, add "CH4" to your list of isotopic compositions investigated. | |
| | **AC: This has been changed in the revised version of our manuscript.** | Line 9 p 13 |
| 12 | Pg 13, Section 4.3.1. The question mark for group 3 in Figure 8 needs an explanation– i.e. what does the "question mark" represent? | |
| | **AC: This question mark has been removed from Fig. 8 in the revised version of our manuscript to avoid confusion.** | See revised Fig. 8 |
| 13 | Pg 14, Line 20+, This part needs clarification because you go on to say, and show in Figure 11, that there are some samples with evidence of methane oxidation, whereas you say here that there is no evidence of oxidation. Be more specific here, or simply remove statement and save for later when you discuss the higher d13C-CH4 values. | |
| | **AC: This statement has been deleted in the revised version of our manuscript.** | Line 27 p 14 |
| 14 | Pg 15, Line 27, "has not yet been oxidized" – be more specific: e.g. there is no evidence of methane oxidation because the elevated d13C-CH4 values are not associated with low d13C-DIC values, as expected for methane oxidation. | |
| | **AC: As suggested, this sentence has been added in the revised version of our manuscript as "**since there is no evidence of methane oxidation such as low $\delta^{13}C_{DIC}$ values as expected for methane oxidation"**.** | Line 4-5 p 16 |
| 15 | Pg 16, Line 8, change "imparting a d13C value" to "imparting a relatively high d13C value". | |
| | **AC: This modification has been integrated in the revised version of our manuscript.** | Line 15 p 16 |
| 16 | Pg 16, Line 10, "It is also possible that post-sampling degradation of low-methane samples occurred." Be more specific. What could have happened (physically) and how might that have changed the isotopic values? | |
| | **AC: The statement "post-sampling degradation of low-methane samples occurred" refers to potential slow diffusive gas loss from sampling containers resulting in $^{13}$C enrichment in the residual methane. This has been clarified in the revised version of the manuscript.** | Line 18-19 p 16 |
| 17 | Pg 17, Line 23, change "interpretation of gas composition" to "interpretation of natural gas composition." | |
| | **AC: This correction has been added to the revised version of our manuscript.** | Line 1 p 18 |
| 18 | Pg 18, Line 3: add an "and" between "d13C, dD". | |
| | **AC: This has been corrected in the revised version of our manuscript.** | $\delta$D removed from the discussion |

| | | in agreement with the comment #21 |
|---|---|---|
| 19 | Pg 18, Line 7, be more specific about this statement. For example, could add text at the end of the sentence: "such as the introduction of deeply-sourced thermogenic gases into shallow aquifers." | |
| | **AC: We have followed the advice of the reviewer and have added this statement to the revised version of our manuscript.** | Line 14-15 p 18 |
| 20 | Figure 3, part b, It's not clear what the "(C(-4)) becomes stable" label represents. Need to explain in the figure caption and/or text. | |
| | **AC: This note has been removed from Fig. 3 in the revised version of our manuscript to avoid confusion.** | See revised Fig. 3 |
| 21 | Figure 11, part b, there wasn't much (if any?) discussion of the d2H-CH4 values in the text - add. BTW - I'm no longer using this plot in my own research because I've found that it is misleading; the dD-CH4 values are low in these western, higher latitude regions not because of a shift in metabolic pathway to "methyl type fermentation", but rather because of isotopic exchange between the 2H in the CH4 and H2O (shown in several studies now). Recent microbial studies from several coalbed methane and black shale systems show that both CO2 reduction and acetoclastic methanogenesis are typically important/present. I have a paper in review on this topic that will hopefully be published soon. In the meantime, see Bates et al. (2011) Chemical Geology for a reference. | |
| | **AC: We fully agree with the reviewer's reasoning especially since the $\delta^2H$ values of groundwater in Alberta are very negative. We therefore have removed Figure 11b and the associated discussion in the text.** | Fig. 11b and associated discussion removed from the text |
| 22 | Figure 12, part b, need to subscript "CH4" in the y-axis label. | |
| | **AC: This subscript has been added in the y-axis label of Fig. 12 in the revised version of our manuscript.** | See revised Fig. 12 |

**From reviewer #2**

| # | Authors Comments (AC) | |
|---|---|---|
| 1 | We have changed the units in Figures 6, 7 and 9 to mmol/L as suggested; | See Revised Figs. 6, 7 and 9 |
| 2 | The ordinate label for Figure 12 has been refined by using mmol/L as a unit for sulfate concentration and by changing the number increments on the x-axis of both graphs ( $\Delta$ x = 1 mmol/L for the sulfate concentration axis and $\Delta$ x = 2 ‰ for $\delta^{13}C_{DIC}$ axis). | See revised Fig. 12 |
| 3 | Table 2 has been changed to show significant digits consistent with the stated measurement uncertainty of the respective parameters; | See Table 2 |

**Additional**

| **Authors Comments (AC)** | |
|---|---|
| We have also made additional minor changes in other figures to ensure consistency with respect to units as used throughout the text. | See revised Fig. 3a; Fig. 4c; Fig. 10a |

[revised manuscript text omitted]